# Identification of a novel toxicophore in anti-cancer chemotherapeutics that targets mitochondrial respiratory complex I

Zoe A Stephenson[1], Robert F Harvey[1†], Kenneth R Pryde[1†], Sarah Mistry[2], Rachel E Hardy[1], Riccardo Serreli[3], Injae Chung[3], Timothy EH Allen[1], Mark Stoneley[1], Marion MacFarlane[1], Peter M Fischer[2], Judy Hirst[3]*, Barrie Kellam[2]*, Anne E Willis[1]*

[1]MRC Toxicology Unit, University of Cambridge, Cambridge, United Kingdom; [2]School of Pharmacy, Biodiscovery Institute, University of Nottingham, Nottingham, United Kingdom; [3]MRC Mitochondrial Biology Unit, University of Cambridge, Cambridge, United Kingdom

**Abstract** Disruption of mitochondrial function selectively targets tumour cells that are dependent on oxidative phosphorylation. However, due to their high energy demands, cardiac cells are disproportionately targeted by mitochondrial toxins resulting in a loss of cardiac function. An analysis of the effects of mubritinib on cardiac cells showed that this drug did not inhibit HER2 as reported, but directly inhibits mitochondrial respiratory complex I, reducing cardiac-cell beat rate, with prolonged exposure resulting in cell death. We used a library of chemical variants of mubritinib and showed that modifying the 1*H*-1,2,3-triazole altered complex I inhibition, identifying the heterocyclic 1,3-nitrogen motif as the toxicophore. The same toxicophore is present in a second anti-cancer therapeutic carboxyamidotriazole (CAI) and we demonstrate that CAI also functions through complex I inhibition, mediated by the toxicophore. Complex I inhibition is directly linked to anti-cancer cell activity, with toxicophore modification ablating the desired effects of these compounds on cancer cell proliferation and apoptosis.

*For correspondence:
jh@mrc-mbu.cam.ac.uk (JH);
barrie.kellam@nottingham.ac.uk
(BK);
aew80@cam.ac.uk (AEW)

†These authors contributed equally to this work

Competing interests: The authors declare that no competing interests exist.

## Introduction

The pharmaceutical industry must deliver safe and effective medicines while simultaneously limiting the costs associated with drug development, and a central part of this process is de-risking potential safety liabilities at an early stage (*Morgan et al., 2018*). In many cases, lack of mechanistic understanding about how compounds cause toxicity hampers predictions of adverse drug reactions (ADRs), and more information about the specific substructures of drug molecules that cause ADRs is required. Such information can then be used to populate machine learning algorithms to generate adverse outcome pathways (AOPs) that predict likely outcomes (*Dey et al., 2018*) from off-target toxicities and that can be used in early phase drug design (*Allen et al., 2018*).

It is widely accepted that disruption of mitochondrial function is a common cause of ADRs and it has been proposed that mitochondrial toxicity, which has a major role in idiosyncratic drug toxicity (*Uetrecht and Naisbitt, 2013*), is responsible for up to 50% of post-market drug withdrawals (*Will and Dykens, 2014*; *Dykens and Will, 2007*). Mitochondrial toxins often have a differential effect on tissue function due to organ-specific variations in the mitochondrial proteome and its function (*Johnson et al., 2007a*); primarily biosynthetic and metabolic in liver, and energy production in muscle (*Johnson et al., 2007b*; *Calvo and Mootha, 2010*). For example, cardiomyocytes, which

**eLife digest** The pharmaceutical industry needs to make safe and effective drugs. At the same time this industry is under pressure to keep the costs of developing these drugs at an acceptable level. Drugs work by interacting with and typically blocking a specific target, such as a protein in a particular type of cell. Sometimes, however, drugs also bind other unexpected targets. These "off-target" effects can be the reason for a drug's toxicity, and it is important – both for the benefit of patients and the money that can be saved when developing drugs – to identify how drugs cause toxic side effects. The earlier researchers detect off-target effects, the better.

Recent data has suggested that an anti-cancer drug called mubritinib has off-target effects on the compartments within cells that provide the cell with most of their energy, the mitochondria. This drug's intended target is a protein called HER2, which is found in large amounts on the surfaces of some breast cancer cells. Yet if mubritinib has this off-target effect on mitochondria, it may be harmful to other cells including heart cells because the heart is an organ that needs a large amount of energy from its mitochondria.

Stephenson et al. have now performed experiments to show that mubritinib does not actually interact with HER2 at all, but only targets mitochondria. The effect of mubritinib as an anti-cancer drug is therefore only due to its activity against mitochondria. Digging deeper into the chemistry revealed the small parts of its chemical structure that was responsible for mubritinib's toxicity against heart cells, the so-called toxic substructure. Another anti-cancer drug called carboxyamidotriazole also has the same toxic substructure. Carboxyamidotriazole is supposed to stop cells from taking up calcium ions, but a final set of experiments demonstrated that this drug also only acts by inhibiting mitochondria.

Often there is not enough information about many drugs' substructures, meaning off-target effects and toxicities cannot be predicted. The pharmaceutical industry will now be able to benefit from this new knowledge about the toxic substructures within some drugs. This research may also help patients who take mubritinib or carboxyamidotriazole, because their doctors will have to check for side effects on the heart more carefully.

have high energy requirements and are rich in mitochondria (*El-Hattab and Scaglia, 2016*), are particularly sensitive to mitochondrial toxins that alter ATP production (*Kolwicz et al., 2013*). Mitochondria are central to many cell-wide processes so mitochondrial toxicity affects bioenergetics, metabolism, signalling and oxidative stress (*Meyer et al., 2018*) in addition to impacting stemness, differentiation and apoptosis (*Guerra et al., 2017*). Due to these pleiotropic roles, 'off-target' drug toxicity resulting in impairment of mitochondrial function can easily be misattributed to other targets and cellular processes.

Given the relative paucity of mechanistic knowledge relating specific drug substructures to ADRs we carried out a detailed chemical dissection of mubritinib, which was reported to act as a tyrosine kinase (HER2) inhibitor and which has been trialled as a treatment for a range of cancers (*Nagasawa et al., 2006*; *Ouchida et al., 2018*). However, mubritinib has also been shown to affect energy status (*Baccelli et al., 2019*; *Sridhar et al., 2003*), and it was recently demonstrated to have an 'off-target' effect on mitochondrial function through inhibition of respiratory complex I in cells derived from patients with Acute Myeloid Leukaemia (AML), leading to a new therapeutic option (*Baccelli et al., 2019*).

Here, we show that mubritinib does not bind directly to HER2 and that its inhibitory effect on complex I negatively impacts on cardiomyocyte function, an important clinical consideration. Through the use of a focussed chemical library, we show that modifying the 1$H$-1,2,3-triazol-1-yl moiety present in mubritinib substantially alters both the inhibition of complex I and the toxicity to cardiomyocytes; identifying a heterocyclic 1,3-nitrogen motif as being key to its complex I inhibitory action. A search of chemical space was then undertaken for the same substructure, and led to the drug carboxyamidotriazole (CAI), which has also been trialled as an anti-cancer agent (*Sridhar et al., 2003*; *Omuro et al., 2018*; *Azad et al., 2009*; *Johnson et al., 2008*). We show that, like mubritinib, CAI does not directly inhibit its reported target (calcium channels) and is also a potent complex I inhibitor. Furthermore, like mubritinib, chemically altering the triazole ring moiety ablated toxicity. In

both cases we show that mitochondrial toxicity is directly linked to anti-cancer activity, as chemical manipulation of the toxicophoric heterocycle ablates the desired effects of both compounds on cancer-cell proliferation and apoptosis. Thus, we have identified a novel toxicophoric motif that is mechanistically linked to an adverse cardiac cell event. Our data demonstrate that caution must be taken when attributing a drug mechanism of action without detailed structure activity relationship (SAR) analysis, since inhibition of mitochondrial function has such cell-wide effects.

## Results

### Mubritinib is not a direct HER2 inhibitor, but instead inhibits ATP production in H9c2 cells and reduces beat rate in cardiomyocytes

Previous data have reported that mubritinib inhibits phosphorylation of HER2 in breast cancer cell lines that express high levels of this receptor (*Nagasawa et al., 2006*). Therefore, the HER2-overexpressing cell line, BT474, was treated with increasing doses of mubritinib and the effect on HER2 phosphorylation was analysed by western blotting, with the specific HER2 inhibitor lapatinib (*Brandão et al., 2018*) as a positive control (*Figure 1A*, *Figure 1—figure supplement 1A*). Surprisingly, there was only a small decrease in HER2 phosphorylation in the presence of mubritinib, in contrast to lapatinib. Furthermore, recent data have further shown that mubritinib, in common with known inhibitors of mitochondrial respiratory complex I (*Sica et al., 2020*), alters the phosphorylation status of proteins that sense changes in energy status and stimulate cellular proliferation, such as mTOR (*Leibovitch and Topisirovic, 2018*). In agreement with these data, we show that treatment of cells with mubritinib (*Figure 1B*) alters the phosphorylation status of proteins downstream of the energy sensor AMPK (e.g. acetyl-CoA carboxylase) and impacts on mTOR signalling (e.g. phosphorylation of RPS6, *Figure 1—figure supplement 1B*). Moreover, similar effects were also observed with treatment by the known mitochondrial inhibitors, rotenone and antimycin A (*Figure 1B* and *Figure 1—figure supplement 1Ci and ii*). In contrast, lapatinib, which is a specific HER2 inhibitor, blocks signalling downstream from mTOR with minimal effect on ACC phosphorylation (*Figure 1B*, *Figure 1—figure supplement 1Ci and ii*), but a large effect on RPS6 (*Figure 1—figure supplement 1Ci and ii*). An inactive mubritinib analogue (compound **5**, see below) was used as a negative control. An in vitro tyrosine kinase activity assay was then carried out with increasing concentrations of mubritinib, in which recombinant human HER2 was incubated with radioactively labelled $^{32}$P-ATP. Mubritinib did not decrease $^{32}$P incorporation, even at 10 μM (*Figure 1C and D*), demonstrating that it does not inhibit HER2 phosphorylation. Furthermore, consistent with our data (*Figure 1E*), mubritinib has been reported to display no activity against almost 300 other kinases screened (*Anastassiadis et al., 2011*).

Because mubritinib affects signalling pathways associated with a decrease in cellular energy, and because inhibiting mitochondrial respiration could have particularly deleterious effects on tissues with high energy demand such as the heart, we tested the effect of mubritinib on ATP production in H9c2 cardiomyoblasts (*Kimes and Brandt, 1976*) and human embryonic stem cell derived cardiomyocytes (hESC-CM, GE Healthcare) using the glucose/galactose system. In galactose-containing media cells are predominantly reliant on mitochondria for the production of cellular ATP, allowing mitochondrial liabilities that are masked in glucose to be revealed (*Marroquin et al., 2007*; *Rana et al., 2011*).

Exposure of H9c2 cells to 2 μM mubritinib for 2 hr in galactose-containing media led to a 50% decrease in ATP levels (*Figure 1F*). Furthermore, prolonged exposure depleted ATP levels to 10% of the control value (*Figure 1F*) and induced cell death (*Figure 1G*). Importantly, inhibition of ATP production in galactose (but not glucose) containing media by mubritinib and other inhibitors of oxidative phosphorylation was also observed in hESC-CMs (*Figure 1H*) and had a profound effect on cell beat rate (*Figure 1I*).

### Structure activity relationships (SARs) determined using a library of derivatives to identify a novel toxicophore in mubritinib

It has been shown recently that mubritinib targets respiratory complex I and inhibits growth of cancer cells from patients with AML, which are highly dependent on oxidative phosphorylation for survival (*Baccelli et al., 2019*). Therefore, to understand likely toxicities associated with such

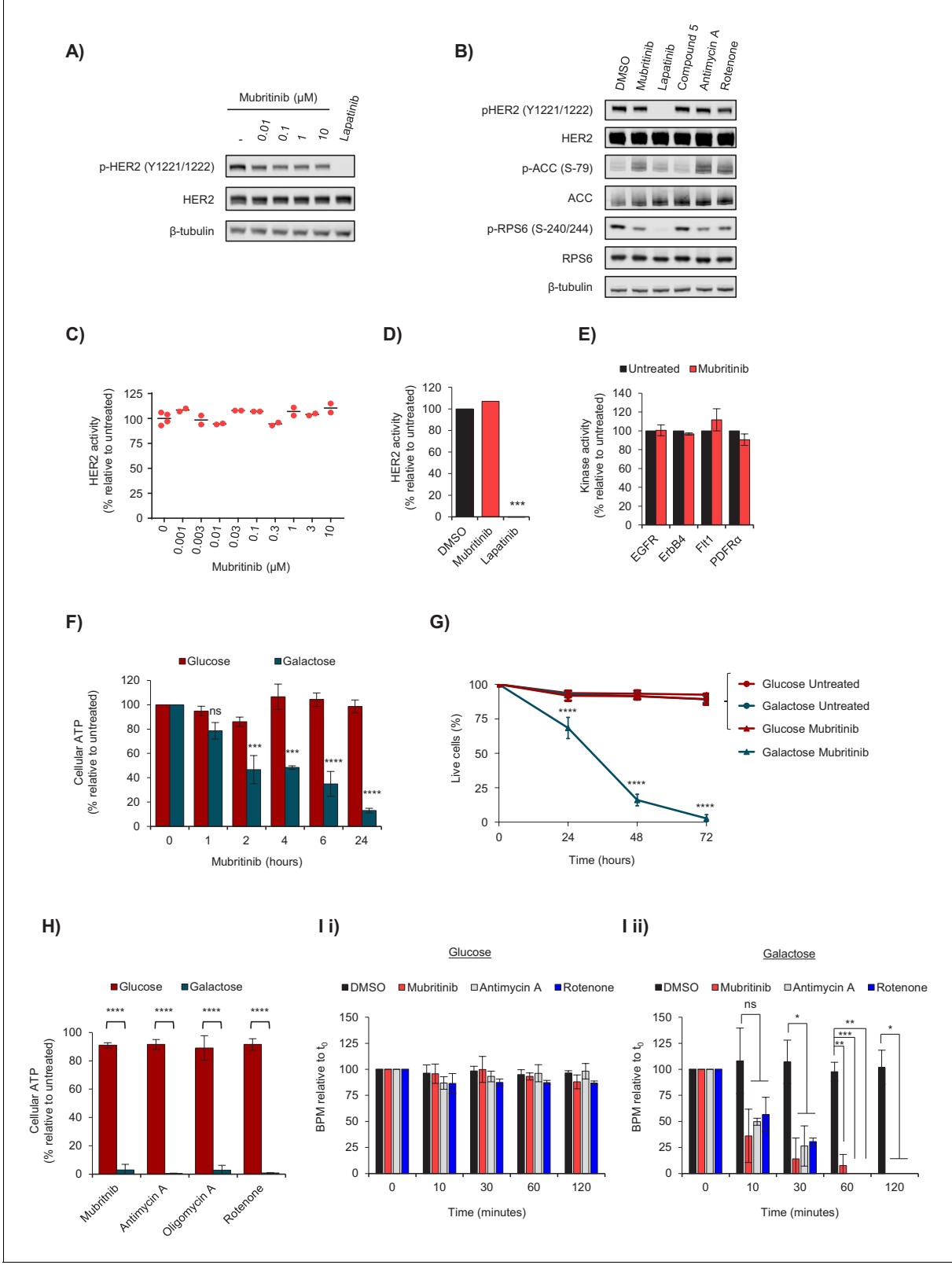

**Figure 1.** Mubritinib does not inhibit HER2, but inhibits ATP production and beat rate of cardiomyocytes. (**A**) Western blot analysis of the HER2-overexpressing cell line, BT474, treated with increasing doses of mubritinib. HER2 activity was assessed with antibodies against phosphorylated HER2 (Y1221/1222). Cells were treated with the clinically used HER2 inhibitor, lapatinib (10 μM), as a positive control. (**B**) Western blot analysis of the HER2-overexpressing cell line, BT474, treated with 1 μM of either mubritinib, lapatinib, compound 5 (inactive mubritinib derivative, see *Figure 2A*), antimycin

*Figure 1 continued on next page*

*Figure 1 continued*

A or rotenone for 2 hr. (C) Radiometric kinase assays were carried out and the effect of mubritinib (at the concentrations shown) on recombinant human HER2 activity was determined by measuring the incorporation of radioactive $^{32}$P-ATP after 15 mins. Activity values are displayed relative to the untreated sample. (D) Radiometric kinase assays were carried out using recombinant human HER2 in the presence of 1 µM mubritinib and lapatinib (DMSO control and lapatinib, n = 3, mubritinib, n = 2). Significance following lapatinib treatment was assessed using the unpaired students t-test (***p<0.001) relative to the DMSO control. (E) Radiometric kinase assays were carried out on recombinant human EGFR, ErbB4, Flt1 and PDFRα in the presence of 2 µM mubritinib. Error bars represent standard deviation (n = 3). (F) A 24 hr time course for loss of ATP from H9c2 cells following treatment with 2 µM mubritinib in media containing either glucose or galactose as the carbon source. Error bars represent standard deviation (n = 3) and significance was assessed using ANOVA with Dunnett's multiple comparisons test (****p<0.0001, ***p<0.001, ns = not significant). (G) H9c2 cells were treated with 2 µM mubritinib in media containing either glucose or galactose as the carbon source and cell viability was assessed over a 72 hr period using DRAQ7 staining and Annexin-V-FITC labelling. Error bars represent standard deviation (n = 3) and significance at each time point was assessed using ANOVA with Tukey's multiple comparisons test (****p<0.0001). (H) hESC-cardiomyocytes (CytivaTM Plus, GE Healthcare) were grown in RPMI-1640 media supplemented with galactose (10 mM) or glucose (11 mM) and treated with 1 µM of mubritinib, or inhibitors of mitochondrial complex III (antimycin A), ATP synthase (oligomycin A) or complex I (rotenone). ATP levels were measured after 2 hr. Error bars represent standard deviation (n = 4) and significance was assessed using the unpaired students t-test (****p<0.0001). (I) hESC-cardiomyocytes (CytivaTM Plus, GE Healthcare) were grown in either galactose (10 mM) or glucose (11 mM) containing media on multi-electrode array plates from which it is possible to assess beat rate. The average beat rates in glucose (i) and galactose (ii) containing media of 52.2 and 30.7 BPM respectively, were set to 100%. Cells were treated with mubritinib (1 µM), antimycin A (1 µM) or rotenone (1 µM). Error bars represent standard deviation (n = 3) and significance was assessed using the unpaired students t-test (*p<0.05, **p<0.01, ****p<0.0001).

The online version of this article includes the following figure supplement(s) for figure 1:

**Figure supplement 1.** Mubritinib targets complex I in cardiomyocytes.

treatments, we investigated whether complex I is similarly affected in cardiomyocytes exposed to mubritinib (**1**), and used a focussed chemical library of mubritinib derivatives (**2-8**) to identify the potential toxicophore (*Figure 2A*). The mubritinib variants were synthesised with modifications in two regions of the molecule; the aryl trifluoromethyl group (compounds **2–4**) and the triazole group (compounds **5–8**).

The oxygen consumption rates (OCR) of H9c2 and hESC-CM cells treated with mubritinib (**1**) were measured to determine whether the decreased ATP content in galactose containing media (*Figure 1F and H*) were due to direct inhibition and/or uncoupling of the respiratory chain (*Felser et al., 2013*). As expected, oligomycin A, which inhibits ATP synthase, had no effect on the rotenone-sensitive OCR in the presence of the uncoupling agent FCCP (carbonyl cyanide-4-(trifluoro-methoxy) phenylhydrazone). However, in H9c2 or hESC-CMs cells exposed to mubritinib the rotenone-sensitive OCR decreased by ~50% (*Figure 2B* and *Figure 2—figure supplement 1A and B*). Taken together, these data suggest that mubritinib (**1**) inhibits the mitochondrial respiratory electron transport chain in cardiomyocytes.

The effects of mubritinib (**1**) on complex I and II linked respiration were then determined in plasma-membrane permeabilised H9c2 cells. First, cells were pre-treated with increasing doses of mubritinib (**1**), or the complex I inhibitor rotenone or the complex III inhibitor antimycin A. As expected, all inhibitor treatments decreased the OCR (*Figure 2C*). Cells were then treated with plasma membrane permeabiliser followed by addition of ADP. Pyruvate/malate and succinate were used to drive respiration from complex I and complex II, respectively and, in untreated cells, they both stimulated the OCR. All three inhibitors inhibited pyruvate/malate-driven respiration but, crucially, the inhibition of only mubritinib and rotenone was alleviated by subsequent treatment with succinate (*Figure 2C*), suggesting that mubritinib is a complex I inhibitor.

Mitochondrial membranes were then used to assess the effect of mubritinib (**1**) on complex I and complex II driven respiration directly. In agreement with the data obtained from cell lines, mubritinib (**1**) showed a dose-dependent decrease in the rate of NADH oxidation and no effect on succinate oxidation (*Figure 2D and E*). The NADH oxidation data were then fit to the standard dose-effect relationship and yielded an IC$_{50}$ value of 19.2 nM (*Figure 2D*).

Purified complex I was then used to confirm inhibition of complex I unambiguously, and to dissect whether mubritinib (**1**) inhibits it at its NADH or ubiquinone binding site. NADH oxidation was coupled to reduction of either the ubiquinone-10 analogue decyclubiquinone (dQ) or to reduction of an artificial electron acceptor (APAD$^+$ or ferricyanide, FeCN) that reoxidises the flavin in the NADH binding site directly (*Birrell et al., 2009*; *Yakovlev and Hirst, 2007*), without the involvement of ubiquinone (*Figure 2E*; *Birrell et al., 2009*). While APAD$^+$ and FeCN reduction were unaffected at 500

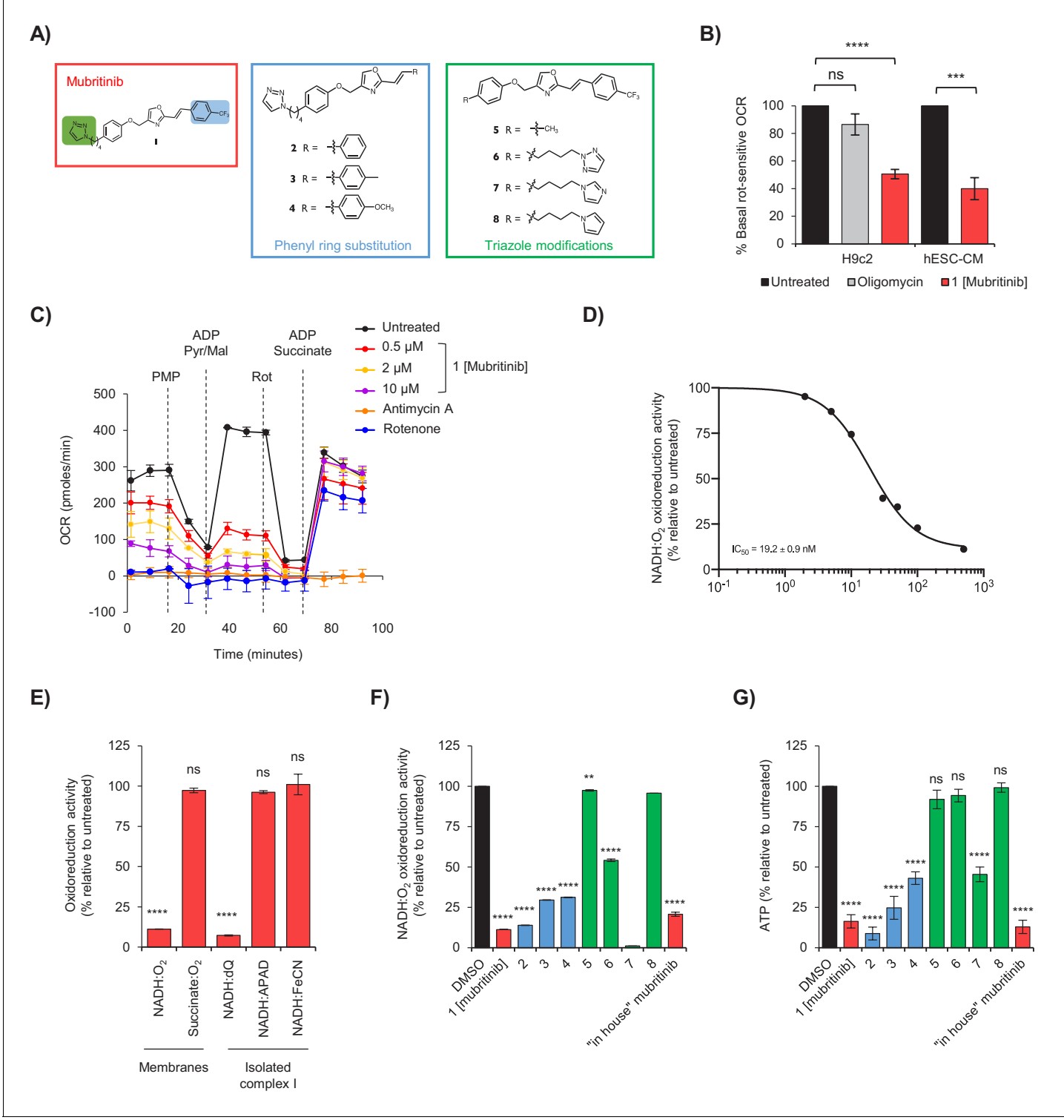

**Figure 2.** Mubritinib is an inhibitor of mitochondrial complex I. (**A**) Variants of mubritinib were synthesised with alterations to the trifluoromethylphenyl group (2, 3 and 4) or the triazole (5, 6, 7 and 8). Mubritinib (1) and mubritinib synthesised 'in house' were used as positive controls. (**B**) Rotenone-sensitive oxygen consumption rates (OCRs) of H9c2 or hESC-CM cells in glucose containing media were measured after the addition of FCCP in the presence of either 1 µM mubritinib or 1 µM oligomycin A. OCR values are represented relative to untreated cells and error bars represent standard deviation (n = 3). Significance was assessed using the unpaired students t-test (*** p = <0.001, **** p = <0.0001, ns = not significant). (**C**) OCR was measured in cells pre-treated with 0.5 µM, 2 µM and 10 µM mubritinib, 1 µM rotenone or 10 µM antimycin A using a Seahorse XF Analyzer. PMP was added to permeabilise the plasma membrane, followed by pyruvate, malate and ADP to drive complex I linked respiration. Then, rotenone was added

*Figure 2 continued on next page*

*Figure 2 continued*

to abolish complex I respiration followed by ADP and succinate to drive complex II linked respiration. Error bars represent standard deviation (n = 3). (D) Mubritinib was incubated with mitochondrial membranes from bovine heart at the concentrations shown, then the rate of NADH was measured spectrophotometrically. Error bars represent standard error of the mean (n = 3). Data were fit using activity (%) = 100/(1 + (IC$_{50}$ / [inhibitor])$^{\text{Hill slope}}$ and yielded an IC$_{50}$ value of 19.2 nM. (E) Relative rates of NADH or succinate oxidation by mitochondrial membranes or complex I isolated from bovine heart using O$_2$, dQ, APAD$^+$, or FeCN as the electron acceptor in the presence of 500 nM mubritinib. Error bars represent standard deviation (n = 3) and significance was assessed using ANOVA with Dunnett's multiple comparisons test (****p<0.0001, ns = not significant). (F) Mubritinib and the variants from (A) were incubated with mitochondrial membranes at 500 nM. The rate of NADH oxidation was measured spectrophotometrically. The activity is expressed relative to the DMSO control, set to 100%. Error bars represent standard deviation (n = 3) and significance was assessed using ANOVA with Dunnett's multiple comparisons test (****p<0.0001, **p<0.01). Activities were inter/extrapolated from measured data points for compounds 7 and 8. (G) H9c2 cells were treated with mubritinib and all compound variants (10 µM) in galactose containing media for 24 hr and ATP levels were measured. Error bars represent standard deviation (n = 3) and significance was assessed using ANOVA with Dunnett's multiple comparisons test (****p<0.0001, ns = not significant).

The online version of this article includes the following figure supplement(s) for figure 2:

**Figure supplement 1.** Mubritinib inhibits OCR in H9c2 and hESC-CM.

**Figure supplement 2.** Complex I inhibition by mubritinib and the synthesised variant compounds.

---

nM mubritinib (**1**), the rate of ubiquinone reduction was essentially abolished (*Figure 2E*). These data confirm that mubritinib inhibits complex I directly by inhibiting ubiquinone reduction, most likely by binding in the ubiquinone-binding site.

The variants of mubritinib were then tested for their ability to inhibit complex I in mitochondrial membranes (*Figure 2F* and *Figure 2—figure supplement 2*). Compounds **2–4** (*Figure 2A*), which have the same *N*1-linked triazole moiety as mubritinib but contain either no substituent (**2**) or a mild (**3**) or strong (**4**) electron donating group in the para-position of the phenyl ring, retain the ability to inhibit complex I, similarly to mubritinib (*Figure 2F*). In contrast, compound **6**, which has the 1,2,3-triazol-1-yl moiety substituted for an isomeric 1,2,3-triazol-2-yl group, is a much weaker inhibitor (*Figure 2F*). Complete removal of the triazole group in **5** and modification of the triazole to an *N*-linked pyrrole in **8**, also resulted in compounds that no longer inhibited NADH oxidation (*Figure 2F*). Most interestingly, modification of the triazole to the *N*-linked imidazole **7** retained the inhibitory activity. These data provide strong evidence that a 1,3-amidine-like motif, housed within the 1*H*-1,2,3-triazol-1-yl substituent, is required for complex I inhibition. The same pattern of inhibition was observed for ATP production in cells grown in media containing galactose (*Figure 2G*). Therefore, inhibition of ATP production by mubritinib results from the inhibition of complex I, and depends strongly upon its 1,2,3-triazol-1-yl moiety and the embedded toxicophore.

## The 1,2,3-triazol-1-yl toxicophore in Carboxyamidotriazole (CAI) inhibits ATP production, mitochondrial function and cell proliferation

Based on the 1*H*-1,2,3-triazol-1-yl moiety being critical for the function of mubritinib we initially searched for other compounds that contain a 1,2,3-triazol-1-yl or similar moiety that might display analogous toxicity profiles. A structural similarity search carried out on the ChEMBL database revealed a range of terminal 1,2,3-triazole-containing drugs as putative complex I inhibitors (*Supplementary file 1*) including carboxyamidotriazole (**9**, *Figure 3A*), which has been trialled widely as an anticancer agent in single and combination therapies to treat glioblastoma, ovarian cancer and non-small cell lung cancer (*Azad et al., 2009*; *Johnson et al., 2008*). Data from a number of studies have suggested that the anti-proliferative and anti-metastatic properties of CAI (**9**) are mediated through the inhibition of non-voltage gated Ca$^{2+}$ channels in non-excitable cells (*Hupe et al., 1990*), which in turn modulates downstream phosphorylation events (*Bauer et al., 2000*), however this has not been demonstrated directly. Calcium binding assays performed here show clearly that there is no significant binding of CAI (**9**) to non-voltage gated Ca$^{2+}$ channels (*Supplementary file 2*). It has also been shown that CAI (**9**) affects mitochondrial calcium import and local calcium clearance, which is essential for the maintenance of capacitative calcium entry (*Mignen et al., 2005*) and it was proposed that this then inhibited oxidative phosphorylation (*Ju et al., 2016*). These data suggest that the toxicophore, in the context of CAI (**9**), might act in a similar way to mubritinib, and that the effects on mitochondrial calcium release could be the secondary effects of complex I inhibition. To explore the effect of the heterocycle in CAI (**9**) on mitochondrial function we generated three

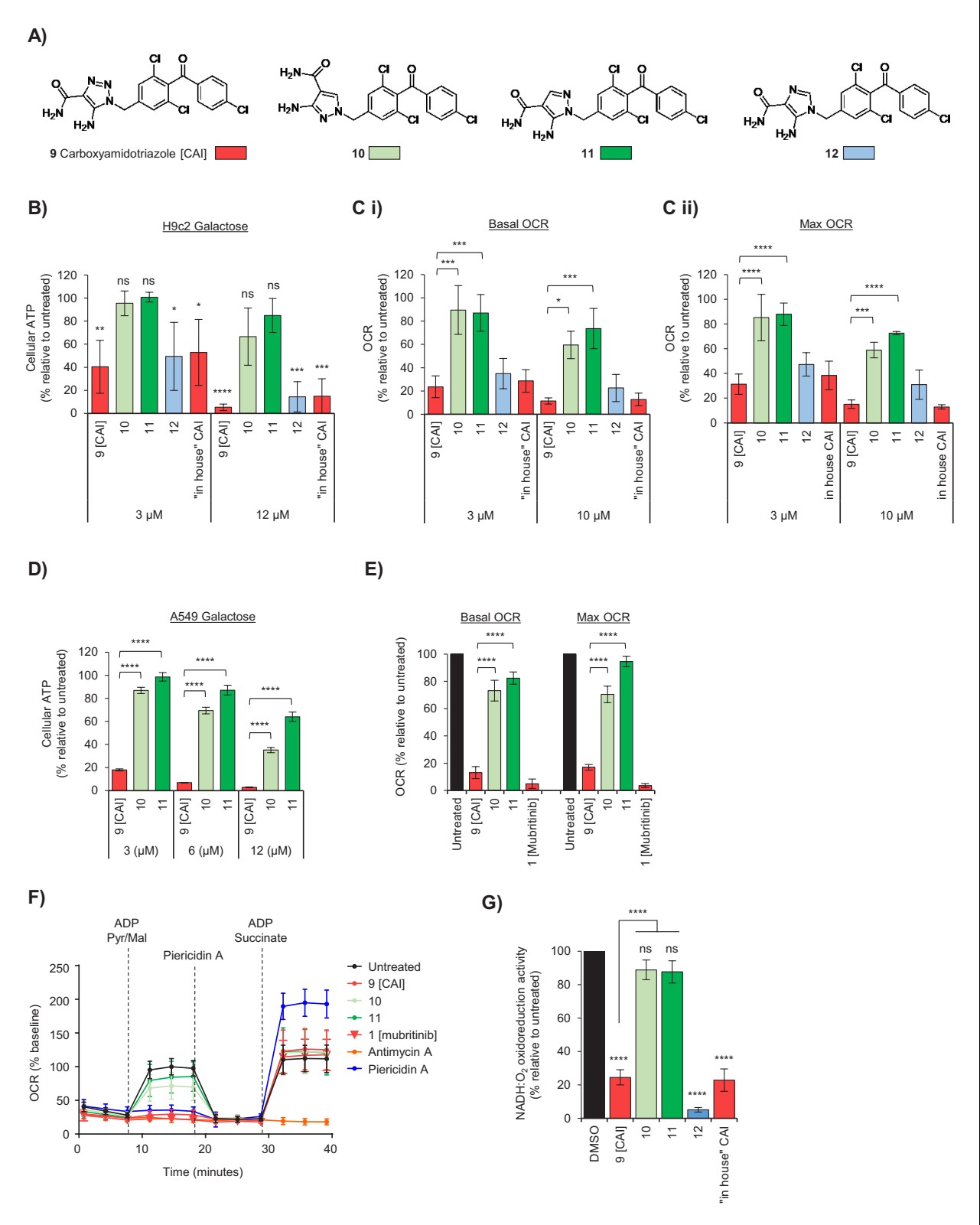

**Figure 3.** The toxicophore present in carboxyamidotriazole inhibits mitochondrial complex I. (**A**) Chemical structure of carboxyamidotriazole (CAI) (**9**) and three variants whereby the core triazole ring was replaced with either a pyrazole (**10** and **11**) or imidazole (**12**). For the pyrazoles, **11** is an analogue of **9** with one of the triazole nitrogens removed, whereas **10** also removes one of the triazole nitrogen atoms whilst additionally shifting the trichlorobenzophenonemethyl moiety to the 2-position equivalent. (**B**) H9c2 cells were treated with 3, 6 or 12 µM of CAI (**9**), **10**, **11**, **12** or the 'in house'

*Figure 3 continued on next page*

*Figure 3 continued*

synthesised CAI in galactose containing media and after 24 hr ATP levels were measured and data shown are relative to the untreated control (n = 3). Significance was assessed using ANOVA with Dunnett's multiple comparisons test (****p<0.0001, ***p<0.001, **p<0.01, *p<0.05, ns = not significant). (C) Basal (i) and maximum (ii) oxygen consumption rates were measured using a Seahorse XF Analyzer in H9c2 cells treated with 3 and 10 µM of either CAI (9), 10, 11, 12 or the 'in house' synthesised CAI. Error bars represent standard deviation (n = 3) and significance was assessed using ANOVA with Tukey's multiple comparisons test (****p<0.0001, ***p<0.001, *p<0.05). (D) A549 cells were treated with 3, 6 or 12 µM of CAI (9) or 10 and 11 in galactose containing media and after 24 hr ATP levels were measured and normalised to the untreated control. Error bars represent standard deviation (n = 3) and significance was assessed using ANOVA with Tukey's multiple comparisons test (****p<0.0001). (E) Basal and maximum oxygen consumption rates were measured using a Seahorse XF Analyzer in A549 cells treated with 5 µM of either CAI (9), 10, 11; or 5 µM mubritinib. Error bars represent standard deviation (n = 3) and significance was assessed using ANOVA with Tukey's multiple comparisons test (****p<0.0001). (F) A549 cells were pre-treated with either 5 µM CAI (9), 10, 11, mubritinib, antimycin A or piericidin A (complex I inhibitor) and the OCR was measured over the times indicated. PMP was added to permeabilise the plasma membranes followed by addition of pyruvate, malate and ADP to drive complex I respiration and OCR determined. Finally, piericidin A was added to abolish complex I respiration followed by ADP and succinate to drive complex II respiration and OCR was again measured. Representative trace from three independent experiments. (G) CAI (9), 10, 11, 12 and the 'in house' synthesised CAI were incubated with mitochondrial membranes at 500 nM. The rate of NADH oxidation was measured spectrophotometrically. The activity is expressed relative to the DMSO control, set to 100%. Error bars represent standard deviation (n = 3) and significance was assessed using ANOVA with Tukey's multiple comparisons test (****p<0.0001, ns = not significant). Activities were interpolated from measured data points for compounds 10, 11, 12 and 'in house' CAI.

The online version of this article includes the following figure supplement(s) for figure 3:

**Figure supplement 1.** The toxicophore in CAI inhibits mitochondrial complex I.
**Figure supplement 2.** Complex I inhibition by mubritinib and the synthesised variant compounds.
**Figure supplement 3.** CAI inhibits signalling pathways responsive to changes in energy status.

variants. Two compounds, which we predicted by analogy with our mubritinib variants that would be inactive, contained isomeric pyrazoles (**10** and **11**). One further compound was synthesised where the triazole ring in CAI was replaced with an imidazole ring (**12**), which we predicted would retain mitochondrial toxicity (*Figure 3A*). The set of compounds was then tested in H9c2 cardiomyoblasts for their effects on oxygen consumption rates and ATP production (*Figure 3B and C*). CAI (**9**) and compound **12** inhibited ATP production in galactose containing media much more strongly than compounds **10** and **11** (*Figure 3B*). Similarly, CAI (**9**) and compound **12** decreased oxygen consumption rates much more effectively than compounds **10** and **11** (*Figure 3C* and *Figure 3—figure supplement 1*).

Given that CAI has been trialled as an anti-cancer therapeutic against lung cancer (*Johnson et al., 2008*), CAI (**9**) and compounds **10** and **11** were tested in the lung cancer derived cell line, A549, to determine whether the effects on mitochondrial function observed in the H9c2 cardiomyoblasts were replicated. The data show that CAI (**9**) inhibits ATP production in A549 cells grown in galactose, whereas **10** and **11** have minimal effect (*Figure 3D*) with no difference observed, as expected, in glucose containing media (*Figure 3—figure supplement 1C*). Moreover, the basal and maximal OCR of cells treated with CAI (**9**) were significantly reduced, with a much smaller decrease observed with compounds **10** and **11** (*Figure 3E* and *Figure 3—figure supplement 1*).

To confirm that CAI inhibited complex I, A549 cells were treated with PMP followed by addition of ADP, pyruvate and malate (*Figure 3F*). In untreated cells, or cells treated with **10** and **11**, there was a large increase in oxygen consumption as expected, however, oxygen consumption was inhibited in cells treated with CAI (**9**), antimycin A or piericidin. Following addition of succinate and ADP there was an increase OCR in CAI (**9**) and piericidin-treated cells (but not in cells treated with antimycin A), strongly suggesting that CAI (**9**) is a complex I inhibitor. Importantly, given that compounds **10** and **11** only have a small effect, in this chemical context the triazolyl toxicophore contributes in a similar manner to mubritinib. To confirm these data, mitochondrial membranes were used to assess the impact of each compound on complex I-driven respiration. As expected, both CAI (**9**) and **12** inhibited complex I-driven respiration whereas compounds **10** and **11** have essentially no effect (*Figure 3G*), as reflected by the measured $IC_{50}$ values for each compound (*Figure 3—figure supplement 2*). Similar to mubritinib, inhibition of complex I with CAI also affects signalling pathways downstream of the energy sensor AMPK, such as increased ACC phosphorylation and inhibition of mTOR signalling reducing RPS6 phosphorylation (*Figure 3—figure supplement 3*). Importantly

these pathways are unaffected by the non-specific calcium channel inhibitor bebridil hydrochloride or the inactive CAI variant compound **11**.

## Inhibition of cell proliferation and cell death by CAI and mubritinib via complex I inhibition is dependent on the toxicophore

To confirm that the toxicophore we have identified is directly linked to the reported anti-proliferative/cancer chemotherapeutic properties of mubritinib (**1**) and CAI (**9**), the degree of cell death following treatment with these compounds was measured in a range of cancer derived cell lines. The cells lines used were representative of AML (HL60), glioblastoma (M059K), lung (A549), osteosarcoma (U-20S), in addition to HeLa cells since these cancers display varying degrees of dependence on glycolysis versus oxidative phosphorylation for energy production and to provide key metabolites required for tumour cell survival (*Molina et al., 2018*; *Shi et al., 2019*). Cells were grown in galactose or glucose containing media in the presence of mubritinib (**1**), CAI (**9**), or a corresponding inactive variant, **5** or **11** respectively, and the degree of cell death was measured using DRAQ7 staining and Annexin-V-FITC labelling (*Figure 4* and *Figure 4—figure supplement 1*). The data show that treatment of the AML derived cell line with either CAI (**9**) or mubritinib (**1**) caused cell death in both glucose and galactose, while the analogue compounds which lacked the heterocyclic 1,3-nitrogen motif had no effect (*Figure 4A and B*, *Figure 4—figure supplement 1*). The other cell lines used showed no cell death in the presence of glucose (*Figure 4D and F* and *Figure 4—figure supplement 1D,F and G*), however again there was a correlation between the presence of the toxicophore and cell death in galactose (*Figure 4C and E* and *Figure 4—figure supplement 1C,E and H*). To explore whether the toxicophore had similar effects on cell growth, BT474 (*Figure 4G*) and A549 (*Figure 4H*) cells were grown in glucose and proliferation measured using xCELLigence RTCA DP instrument. Again, the data show a direct correlation between the presence of the 1,2,3-triazole and cell growth inhibition as both mubritinib (**1**) and CAI (**9**) slowed cell growth, whereas the analogues, which lacked the heterocyclic 1,3-nitrogen motif, had reduced or no effect.

Taken together these data establish that the presence of the triazole and its embedded heterocyclic 1,3-nitrogen toxicophore is essential for the parent drug effects on tumour cell growth of otherwise chemically distinct mubritinib and CAI.

## Discussion

Our data show the value of using SARs to probe the molecular signatures that potentially trigger toxicity pathways and, through the identification of a novel toxicophore, have implications for drug development programmes (*Figure 5*). The toxicophore in the context of mubritinib and CAI is the embedded 1,3-nitrogen motif of the 1*H*-1,2,3-triazol-1-yl heterocycle and we have shown that this nitrogen atom disposition appears critical for both mitochondrial toxicities (*Figures 1–4*). In a preliminary screen of compounds that inhibited complex I, we identified two antifungal agents, ketoconazole and terconazole, which also contained the heterocyclic 1,3-nitrogen motif embedded within a 1*H*-1,2,4-triazol-1-yl or 1*H*-imidazol-1-yl substituent respectively (*Supplementary file 3*). Interestingly, we also found that rufinamide (*Supplementary file 1*), which contains a chemically similar core scaffold to CAI, displayed no complex I inhibitory activity (*Serreli, 2018*). However, there are key structural differences between these two drugs and in particular, rufinamide lacks an anilino nitrogen in the 5-position of the 1,2,3-triazole, the *para*-chlorobenzoyl moiety and there is a chloro to fluoro switch with regards the halogen substituents on the *N*-benzyl group (alongside the switch in position from 3,5 to 2,6). These observations are therefore coupled with a difference in the logP of rufinamide (1.3) when compared to CAI (3.1) which indicates a significant decrease in its overall lipophilicity and may therefore reflect a compromised pharmacokinetic-driven target engagement. Which of these differences drive the observed loss in complex I inhibition by rufinamide is a focus of ongoing work.

It is of serious concern that while both mubritinib and CAI are trialled as part of anti-cancer therapies (*Baccelli et al., 2019*), neither directly bind their reported targets, HER2 and $Ca^{2+}$ channels respectively (*Figure 1* and *Supplementary file 2*). Mitochondria play a central role in cell-wide processes in addition to bioenergetics and metabolism by providing a signalling hub that controls stemness, differentiation and apoptosis (*Guerra et al., 2017*), therefore disruption of mitochondrial function can easily be misattributed to other targets, such as receptors involved in cell signalling.

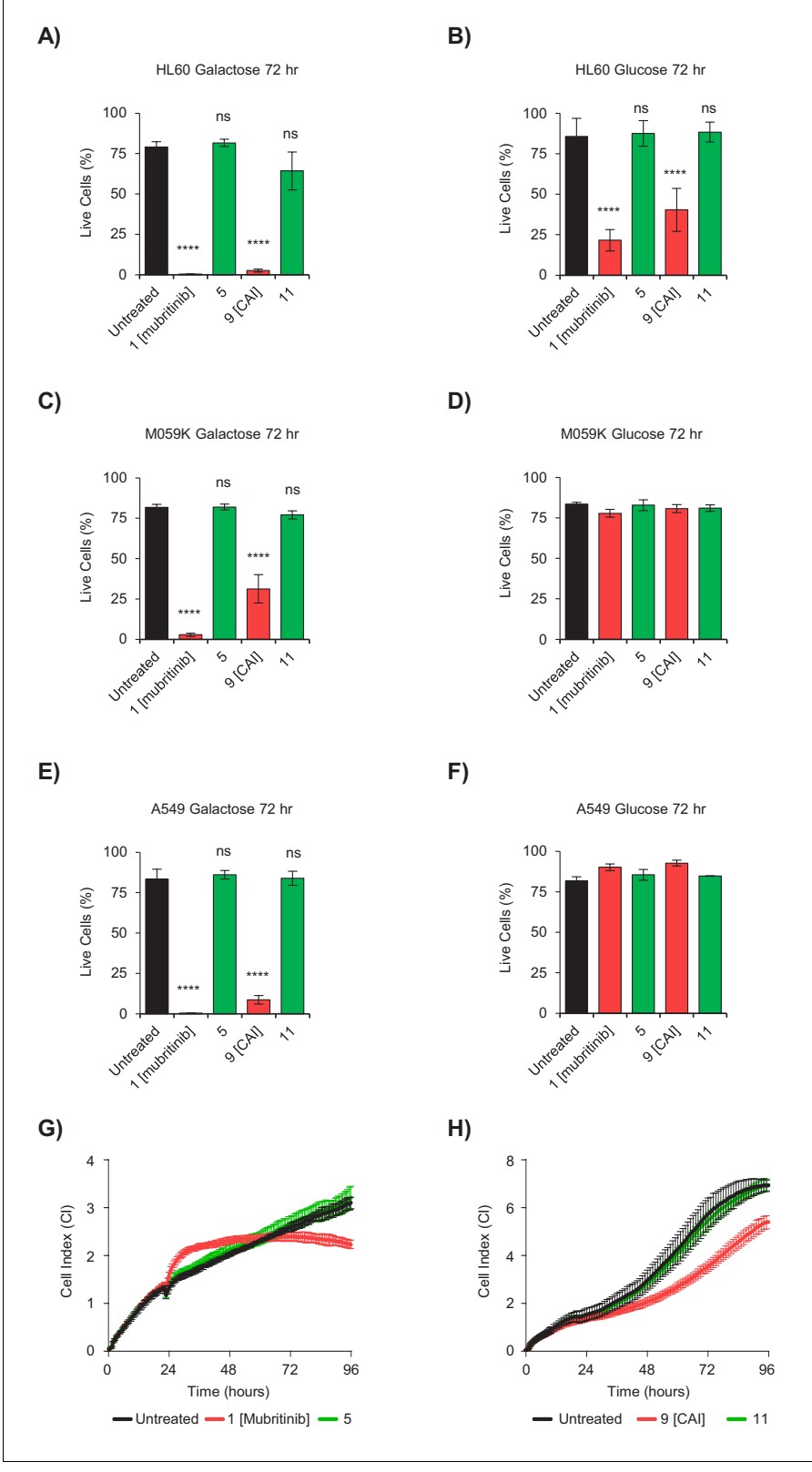

**Figure 4.** The toxicophore present in mubritinib and CAI is required for efficacy as an anti-cancer agent. (**A and B**) HL60 cells grown in media containing galactose (**A**) or glucose (**B**) as an energy source treated with mubritinib (1) (2 µM), CAI (9) (5 µM) or the inactive derivatives 5 (2 µM) or 11 (5 µM) for 72 hr. The percentage of live cells was assessed by DRAQ7 staining and Annexin-V-FITC labelling. Error bars represent standard deviation (n = 3) and

*Figure 4 continued on next page*

*Figure 4 continued*

significance relative to the untreated control sample was assessed using ANOVA with Dunnett's multiple comparisons test (****p<0.0001, ns = not significant). (**C and D**) M059K cells grown in media containing galactose (**C**) or glucose (**D**) as an energy source treated with mubritinib (1) (2 µM), CAI (9) (5 µM) or the inactive derivatives 5 (2 µM) or 11 (5 µM) for 72 hr. The percentage of live cells was assessed by DRAQ7 staining and Annexin-V-FITC labelling. Error bars represent standard deviation (n = 3) and significance relative to the untreated control sample was assessed using ANOVA with Dunnett's multiple comparisons test (****p<0.0001, ns = not significant). (**E and F**) A549 cells grown in media containing galactose (**E**) or glucose (**F**) as an energy source treated with mubritinib (1) (2 µM), CAI (9) (5 µM) or the inactive derivatives 5 (2 µM) or 11 (5 µM) for 72 hr. The percentage of live cells was assessed by DRAQ7 staining and Annexin-V-FITC labelling. Error bars represent standard deviation (n = 3) and significance relative to the untreated control sample was assessed using ANOVA with Dunnett's multiple comparisons test (****p<0.0001, ns = not significant). (**G**) Cell proliferation profiles from xCELLigence RTCA DP instrument. BT474 cells grown in glucose containing media were seeded in an E-plate 16 and after 24 hr were treated with either 5 µM mubritinib (1) or the derivative compound five which contains a modified triazole ring. The cell index was monitored for 96 hr to determine cell proliferation rates. Error bars represent standard deviation (n = 3). (**H**) Cell proliferation profiles from xCELLigence RTCA DP instrument. A549 cells grown in glucose containing media were seeded in an E-plate 16 and after 20 hr were treated with either 5 µM CAI (9) or the derivative compound 11 which contains a modified triazole ring. The cell index was monitored for 120 hr to determine cell proliferation rates. Error bars represent standard deviation (n = 3).

The online version of this article includes the following figure supplement(s) for figure 4:

**Figure supplement 1.** CAI and mubritinib inhibit cell growth and induce apoptosis in glycolytic-deficient tumour cell lines.

---

Cardiac cells are especially sensitive to mitochondrial toxicants that alter ATP production and our data suggest that both mubritinib and CAI have the potential to affect these cell types in situations when glucose is limiting (*Figures 1*, *2* and *4*). It would therefore be important to monitor patients treated with such agents for changes in cardiac function during future clinical trials of CAI or mubritinib.

In terms of cancer treatment, mubritinib has been used to chemo-sensitise tumour cells to other cytotoxic agents. For example, in combination with AC220 (quizartinb), mubritinib reduces the viability of ovarian derived cell lines (*Ouchida et al., 2018*). However, sole inhibition of complex I is also a viable treatment option for some cancer types. Thus while drugs that target the coordinated upregulation of glycolysis that is often associated with tumorigenesis are being developed, several studies have shown that many cancer cell subpopulations are particularly dependent upon OXPHOS for bioenergetic and biosynthetic processes (*Vazquez et al., 2013*; *Viale et al., 2014*; *Birsoy et al., 2015*). Compounds that target the mitochondria, either complex I including mubritinib (*Baccelli et al., 2019*) and IACS-010759 (*Molina et al., 2018*), or the ATP synthase such as Giboxin (*Shi et al., 2019*), have been shown to have efficacy in glycolysis–deficient tumour cells derived from patients with AML and glioblastoma (*Baccelli et al., 2019*; *Bauer et al., 2000*; *Shi et al., 2019*). Interestingly, IACS-010759 (*Bauer et al., 2000*) contains a 1*H*-1,2,4-triazole suggesting a similar mode of action of this drug since this heterocycle also possesses the amidine-like nitrogen substitution pattern.

Since the efficacy of both mubritinib and CAI (*Figure 4*) is dependent upon an amidine-like nitrogen substructure our data suggest that new drug development programmes using this substructure within the correct chemical context could be employed to devise therapies for glycolysis deficient tumours, providing that cardiac liabilities are assessed and evaluated.

Therefore, a more detailed examination of potential molecular recognition of these and other substructures by complex I is ongoing within our laboratories, coupled with consideration of target access from a pharmacokinetic perspective to allow repurposing of a number of drugs for their anticancer properties.

## Materials and methods

### Cell culture

All cell lines were obtained from ATCC, except hESC-cardiomyocytes which were obtained from GE healthcare (Cytiva Plus, GE Healthcare). Glucose containing media consists of glucose-containing

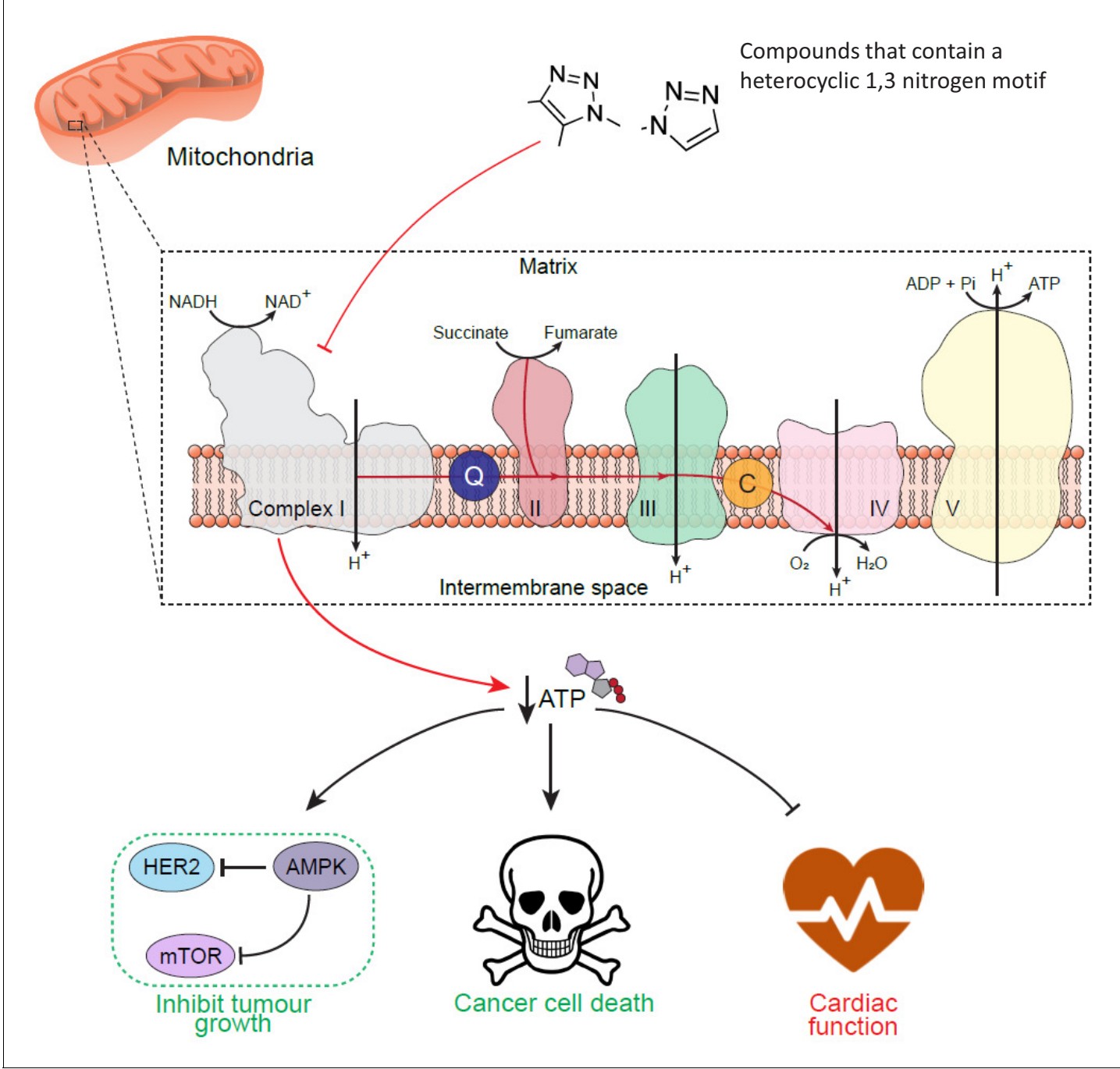

**Figure 5.** Schematic diagram to show the cell-wide effect of ETC complex I inhibition by the toxicophore. Chemical inhibition of mitochondrial respiratory complex I with mubritinib and CAI leads to a decrease in cellular ATP and the subsequent activation of the energy sensor AMPK. Importantly, AMPK has been shown to phosphorylate and inhibit HER2 (*Jhaveri et al., 2015*) suggesting how mubritinib has been misattributed as a HER2 inhibitor, as well to negatively regulate protein synthesis via the mTOR axis to inhibit tumour cell growth. Moreover, cancer cells that are dependent on oxidative phosphorylation for ATP production, such as in AML, will be more sensitive to compounds that inhibit complex I. However, the decrease in ATP levels following treatment with these compounds will also have a particularly toxic effect on tissues with a high energy demand, such as cardiac tissue, and thus impact on heart function.

DMEM (Life Technologies, Gibco 41966) supplemented with 10% FBS. Galactose containing media consists of glucose-free DMEM (Life Technologies, Gibco) supplemented with 10 mM galactose, 2 mM L-glutamine, 1 mM sodium pyruvate and 10% FBS. Prior to treatments, cells were grown in their respective media overnight. hESC-cardiomyocytes were cultured in RPMI-1640 media supplemented with either glucose (11 mM) or galactose (10 mM). All cell lines were routinely tested to ensure that they were mycoplasma free.

## Western blot analyses

Whole cell lysates were prepared in lysis buffer (50 mM Tris pH 7.5, 150 mM sodium chloride, 1% Triton X-100, 0.1% SDS, 0.5% sodium deoxycholate, 1X Roche protease inhibitor cocktail and 1X Roche PhosStop phosphatase inhibitor cocktail). Protein amount was quantified using Pierce BCA protein assay kit (ThermoFisher scientific) and 25 µg protein was separated using SDS-PAGE and transferred to PVDF membranes. Primary antibodies used: phospho-HER2 (Y1221/1222) (CST, #2243), HER2 (CST, #2165), phospho-ACC (S-79) (CST, #3661), ACC (CST, #3676), phospho-RPS6 (S240/244) (CST, #2215), RPS6 (CST, #2217), β-tubulin (CST, #2146). Secondary antibodies used: IR-dye labelled α-rabbit (CST, #5366S). Fluorescent signal was detected using LI-COR Odyssey imaging system (LI-COR biosciences) and images analysed with LI-COR image studio software (package version 5.2.5).

## Measurement of ATP content

H9c2 rat cardiomyoblast cells and hESC-cardiomyocytes were seeded in 96 well plates at a density of $7 \times 10^3$ and $3.6 \times 10^5$ cells per well respectively, whereas A549 cells were seeded at a density of $1 \times 10^4$ cells per well. ATP concentrations were measured using the Promega Cell Titer Glo assay according to manufacturer's protocol.

## Measurement of bioenergetics in live cells

An Agilent XF Seahorse Analyzer was used to measure respiration and glycolysis in intact cells in real time. Cells were seeded in Seahorse Biosciences XF24 plates. H9c2, hESC-CM (Cytiva Plus, GE Healthcare) and were seeded at $9 \times 10^4$, and $6 \times 10^4$ cells per well (coated with 25 µl fibronectin prior to seeding). Prior to the assay, medium was replaced with DMEM, containing either glucose or galactose. Drugs were dissolved in DMSO and injected from pre-loaded ports pneumatically.

## Measurement of mitochondrial respiration in permeabilised cells

Cholesterol-dependent permeabilser XF Plasma Membrane Permeabiliser (PMP) (Seahorse Biosciences), was used as described (*Salabei et al., 2014*). Mitochondrial assay buffer or mannitol and sucrose buffer (MAS, pH 7.2) (220 mM mannitol, 70 mM sucrose, 10 mM $KH_2PO_4$, 5 mM $MgCl_2$, 2 mM HEPES, 1 mM EGTA, 4 mg/ml BSA) was used. Cells were seeded at $5 \times 10^4$ per well in XF Cell Culture plates in standard culture medium. Cells were washed once with MAS before addition of pre-warmed MAS at a final volume of 675 µl. Pyruvate (5 mM), malate (2.5 mM) and ADP (1 mM) were added for measurement of complex I-driven respiration and succinate (10 mM) and ADP (1 mM) for complex II-driven respiration.

## Preparation of proteins and mitochondrial membranes

Mitochondrial membranes and complex I were prepared from bovine (*Bos taurus*) heart mitochondria as described previously (*Blaza et al., 2014*; *Jones et al., 2016*).

## NADH oxidation assays by mitochondrial membranes and complex I

Assays were performed in 10 mM Tris-SO4 (pH 7.5), 250 mM sucrose at 32°C. For measurement of $NADH:O_2$ oxidoreduction, 20–30 µg mL$^{-1}$ of membranes, 1 or 3 µM horse heart cytochrome *c*, and 120 or 200 µM NADH were added and the absorbance of NADH measured at 340–380 nm using linear regression, once steady-state was reached. Succinate oxidation was determined using a coupled enzymatic assay (*Jones and Hirst, 2013*) in the presence of 5 mM succinate. NADH:decylubiquinone (dQ), NADH:3-acetylpyridine adenine dinucleotide (APAD$^+$), and NADH:ferricyanide (FeCN) oxidoreduction by complex I were measured using 0.5 µg mL$^{-1}$ complex I, 100 µM NADH, 0.075% soy bean

asolectin (Avanti Polar Lipids), 0.075% 3-[(3-Cholamidopropyl)dimethylammonio]−1-propanesulfonate (CHAPS, Merck Chemicals Ltd.) and 100 μM dQ, 500 μM APAD$^+$ or 1 mM FeCN, respectively.

## Quantification of cardiac cell function in hESC-CM using multi-electrode arrays (MEA)

MEA plates (Axion Biosystems, M768-KAP-48) contain 48 wells, each with 16 electrodes. hESC-CM were grown on these plates and incubated with the doses of mubritinib shown and recordings were taken over a time course up to 72 hr. AxIS (Version 2.0.2.9) cardiac beat detector which uses an inflection search algorithm was used to detect changes in cardiac action potential.

## Apoptosis and cell death analysis

For analysis of cell death and apoptosis, cells were harvested and FITC-conjugated Annexin-V antibody and far-red DNA stain DRAQ7 were added to pellets resuspended in Annexin-V Binding Buffer (BD Pharmingen). Samples were analysed by flow cytometry using the FITC and APC channels.

## Assays for ion channel binding

Ion channel binding assays were carried out by Eurofins Panlabs Discovery Services Taiwan Ltd. The binding affinity of compounds to ion channels was measured using an in vitro radioligand binding assay ([$^3$H] 1,4,5-IP3) in rat cerebellum, following incubation at 25°C for 10 min. All data are displayed show a percentage binding of each compound relative to control 1,4,5-IP3.

## Assays for activity against HER2

A radiometric kinase assay was carried out by Eurofins Ltd. In brief, the effect of mubritinib on recombinant human HER2 activity was determined by measuring the incorporation of radioactive $^{32}$P-ATP using concentrations from 10 nM to 10 μM. Activity values represent the percentage relative to the positive control. Mubritinib was also tested for activity against EGFR, ErbB4, Flt1 and PDGRα at a concentration of 1 μM. Lapatinib was tested against HER2 as a positive control. The experiments were carried out in triplicate and the counts per minute (CPMs) were normalised to the control.

## Cell proliferation assays

Cell proliferation assays were performed in a standard $CO_2$ incubator using the xCELLigence RTCA DP instrument (ACEA Biosciences) according to manufacturer's instructions. Microelectrode sensor arrays embedded on the base of the E-plate 16 (ACEA Biosciences) measure changes in impedance as cells attach and proliferate, enabling label free quantification of cell proliferation. A549 or BT474 cells were seeded on an E-plate 16 in a total volume of 150 μl media. Cells were allowed to attach to the plate and enter log phase growth (~20 hr) before treatment with indicated compounds of interest. Cell proliferation was monitored for 96 hr post treatment and all treatments were performed in at least technical duplicate and biological triplicate.

## Chemical synthesis

For synthesis of mubritinib and CAI variants see supplementary material.

## Acknowledgements

This work was funded by the Medical Research Council (MC_UU_000/RG94521 and PUAG015 to AEW and MC_U105663141 and MC_UU_00015/2 to JH) and by an ITTP studentship to ZAS. Thanks to Ryan Mordue for drawing *Figure 5*.

## Additional information

### Funding

| Funder | Grant reference number | Author |
|---|---|---|
| Medical Research Council | MC_UU_000 /RG94521 | Zoe A Stephenson<br>Robert F Harvey<br>Kenneth Pryde<br>Anne E Willis |
| Medical Research Council | PUAG015 | Anne E Willis |
| Medical Research Council | MC_U105663141 | Judy Hirst |
| Medical Research Council | MC_UU_00015/2 | Judy Hirst |

The funders had no role in study design, data collection and interpretation, or the decision to submit the work for publication.

### Author contributions

Zoe A Stephenson, Formal analysis, Investigation, Methodology; Robert F Harvey, Formal analysis, Investigation, Writing - original draft, Writing - review and editing; Kenneth R Pryde, Formal analysis, Investigation, Writing - review and editing; Sarah Mistry, Investigation, Methodology; Rachel E Hardy, Riccardo Serreli, Injae Chung, Timothy EH Allen, Investigation; Mark Stoneley, Judy Hirst, Conceptualization, Supervision, Writing - review and editing; Marion MacFarlane, Conceptualization, Supervision; Peter M Fischer, Conceptualization; Barrie Kellam, Conceptualization, Supervision, Investigation, Writing - review and editing; Anne E Willis, Conceptualization, Formal analysis, Supervision, Funding acquisition, Writing - original draft, Project administration

### Author ORCIDs

Robert F Harvey ⬤ https://orcid.org/0000-0002-0023-1146
Injae Chung ⬤ https://orcid.org/0000-0002-2902-4677
Judy Hirst ⬤ https://orcid.org/0000-0001-8667-6797
Barrie Kellam ⬤ https://orcid.org/0000-0003-0030-9908
Anne E Willis ⬤ https://orcid.org/0000-0002-1470-8531

### Decision letter and Author response

Decision letter https://doi.org/10.7554/eLife.55845.sa1
Author response https://doi.org/10.7554/eLife.55845.sa2

## Additional files

### Supplementary files

• Supplementary file 1. Identification of compounds which contain a non-fused triazole in a conformation similar to mubritinib. ChEMBL was searched for drugs containing a non-fused triazole, which led to the identification of a number of small molecules that are used clinically either routinely or in trials use e.g. the antibiotic tazobactam, the anti-epileptic drug rufinamide, and the cancer chemotherapeutic carboxyamidotriazole. These all contain the triazole ring, but have differing associated physicochemical properties.

• Supplementary file 2. Ion channel binding assay. The $Ca^{2+}$ ion channel binding assay to test the activity of CAI (9), 10, 11, mubritinib (1) and 6 was performed by Eurofins. The percentage inhibition of ion channel was calculated relative to the positive control (1,4,5-IP3). On the scale used a score of 1 = no binding and a score of 100 = binding. The data show that there is no direct binding of these drugs to the ion channels.

• Supplementary file 3. Ketoconazole, terconazole and rufinamide all contain a heterocyclic 1,3-nitrogen motif. The compounds listed were incubated with mitochondrial membranes and the rate of NADH oxidation was measured spectrophotometrically.

• Transparent reporting form

### Data availability

All data generated or analysed during this study are included in the manuscript and supporting files.

---

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

## Appendix 1

## Synthesis of Mubritinib and Carboxyamidotriazole (CAI) Analogues

### General Chemistry: Materials and Methods

Chemicals and solvents of analytical and HPLC grade were purchased from commercial suppliers and used without further purification. All reactions were carried out at ambient temperature unless otherwise stated. Reactions were monitored by thin-layer chromatography on commercially available silica pre-coated aluminium-backed plates (Merck Kieselgel 60 F254). Visualisation was under UV light (254 nm and 366 nm), followed by staining with ninhydrin or KMnO4 dips. Flash column chromatography was performed using silica gel 60, 230–400 mesh particle size (Sigma Aldrich). NMR spectra were recorded on a Bruker-AV 400. $^1$H spectra were recorded at 400.13 Hz and $^{13}$C NMR spectra at 101.62 Hz. All 13C NMR are 1H broadband decoupled. Solvents used for NMR analysis (reference peaks listed) were CDCl$_3$ supplied by Cambridge Isotope Laboratories Inc, ($\delta$H = 7.26 ppm, $\delta$C = 77.16) and CD$_3$OD supplied by VWR ($\delta$H = 3.31 ppm and $\delta$C = 49.00). Chemical shifts ($\delta$) are recorded in parts per million (ppm) and coupling constants are recorded in Hz. The following abbreviations are used to described signal shapes and multiplicities; singlet (s), doublet (d), triplet (t), quartet (q), broad (br), dd (doublet of doublets), ddd (double doublet of doublets), dtd (double triplet of doublets) and multiplet (m). Spectra were assigned using appropriate COSY and HSQC experiments. Processing of the NMR data was carried out using the NMR software Topspin 3.0. LC-MS spectra were recorded on a Shimadzu UFLCXR system coupled to an Applied Biosystems API2000 and visualised at 254 nm (channel 1) and 220 nm (channel 2). LC-MS was carried out using a Phenomenex Gemini-NX C18 110A, column (50 mm ×2 mm x 3 μm) at a flow rate 0.5 mL/min over a 5 min period (Method A). All high-resolution mass spectra (HRMS) were recorded on a Bruker microTOF mass spectrometer using MS electrospray ionization operating in positive ion mode. RP-HPLC was performed on a Waters 515 LC system and monitored using a Waters 996 photodiode array detector at wavelengths between 190 and 800 nm. Spectra were analysed using Millenium 32 software. Analytical RP-HPLC was performed using a YMC-Pack C8 column (150 mm ×4.6 mm × 5 μm) and a Phenomenex Gemini NX-C18 column (250 mm ×4.6 mm × 5 μm) at a flow rate of 1.0 mL/min. Final products were one single peak and >95% pure. The retention time of the final product is reported using a gradient method of 5–95% solvent B in solvent A over 25 min. (Solvent A = 0.01% formic acid in H$_2$O, solvent B = 0.01% formic acid in CH$_3$CN).

Abbreviations; DCM, dichloromethane; DIAD, diisopropylazodicarboxylate; DMF, N,N-dimethylformamide; THF, tetrahydrofuran.

## Mubritinib Analogues (*Appendix 1—figure 1*)

**Scheme 1**. *Reagents & Conditions*: (a) Tosyl chloride, triethylamine, $CH_2Cl_2$, 0°C → r.t., 20 h; (b) 1*H*-1,2,3-triazole, NaOH, NaI, amyl alcohol, Δ, 5h; (c) pyridine hydrochloride, MW (180 °C) 2 min then 6 min; (d) PPh₃, DIAD, 1*H*-1,2,3-triazole, THF, 19 h; (e) pyridine hydrochloride, MW (180 °C) 3 x 2 min; (f) CBr₄, PPh₃, $CH_2Cl_2$, 18 h; (g) Imidazole, NaH, DMF, 18 h; (h) pyridine hydrochloride, MW (180 °C) 6 min.

**Scheme 2**. *Reagents & Conditions*: (a) (i) (COCl)₂, DMF, THF, (ii) NH₃, H₂O, EtOAc; (b) 1,3-dichloroacetone, toluene, Δ; (c) 4-hydroxytoluene, NaH, DMF; (d) 4-substituted phenol (cmpds **15**, **18**, **22** or **23**), NaH, DMF.

**Appendix 1—figure 1.** Mubritinib analogues.

Scheme 1. Reagents and Conditions: (a) Tosyl chloride, triethylamine, $CH_2Cl_2$, 0°C → r.t., 20 hr; (b) 1H-1,2,3-triazole, NaOH, NaI, amyl alcohol, Δ, 5 hr; (c) pyridine hydrochloride, MW (180 °C) 2 min then 6 min; (d) PPh₃, DIAD, 1H-1,2,3-triazole, THF, 19 hr; (e) pyridine hydrochloride, MW (180 °C) 3 × 2 min; (f) CBr₄, PPh₃, $CH_2Cl_2$, 18 hr; (g) Imidazole, NaH, DMF, 18 hr; (h) pyridine hydrochloride, MW (180 °C) 6 min.

Scheme 2. Reagents and Conditions: (a) (i) (COCl)₂, DMF, THF, (ii) NH₃, H₂O, EtOAc; (b) 1,3-dichloroacetone, toluene, Δ; (c) *4-hydroxytoluene, NaH, DMF;* (d) 4-substituted phenol (cmpds **15**, **18**, **22** or **23**), NaH, DMF.

# 2-(4-(4-methoxyphenyl)butyl)-2*H*-1,2,3-triazole (14)

To 4-(4'-methoxyphenyl)−1-butanol (**13**) (0.20 g, 1.1 mmol, 1.0 eq) in THF (7.0 mL), triphenylphosphine (0.29 g, 1.1 mmol, 1.0 eq) was added. The mixture was stirred for 5 min and DIAD (0.27 g, 1.2 mmol, 1.1 eq) was then added. The solution was stirred for a further 5 min and 1*H*-triazole (84.0 mg, 1.2 mmol, 1.1 eq) was added. After stirring for 19 hr the THF was removed in vacuo. The residue was purified by column chromatography (1:9 EtOAc/pet. ether) to afford a pale-yellow oil (0.127 g, 49%). ¹H NMR (400MHz, CDCl₃): δ = 7.58 (s, 2H), 7.06 (d, *J* = 8.6 Hz), 6.81 (d, *J* = 8.7 Hz), 4.45 (t, *J* = 7.08 Hz, 2H), 3.78 (s, 3H), 2.58 (t, *J* = 7.71 Hz, 2H), 1.98 (quintet, *J* = 7.4 Hz, 2H), 1.64–1.54 (m, 2H). ¹³C NMR (100 MHz, CDCl₃): δ = 157.96, 133.99, 129.37, 113.92, 55.39, 54.79, 34.39, 29.35, 28.54. LC-MS m/z calc. for $C_{13}H_{17}N_3O$ [MH]⁺; 232.1, found; 232.2, $t_R$ = 2.81 min.

## 2-(4-(4-hydroxyphenyl)butyl)-2H-1,2,3-triazole (15)

To a microwave vial **14** (0.51 g, 2.2 mmol, 1.0 eq) and pyridine hydrochloride (1.27 g, 1.1 mmol, 5.0 eq) were added. The mixture was heated to 180°C in the microwave for 2 min. This was repeated twice until all starting material was consumed. The brown solid obtained was partitioned between EtOAc and water, and the organic layer was washed a further two times with water and then dried over $Na_2SO_4$. The solvent was removed to afford a brown oil, which was purified by column chromatography (2:8 EtOAc/pet. ether) to afford the title compound as a dark yellow oil (0.33 g, 69%). [1]H NMR (400MHz, $CDCl_3$): δ = 7.59 (s, 2H), 6.99 (d, $J$ = 8.5 Hz), 6.72(d, $J$ = 8.5 Hz), 5.51 (s, 2H), 4.45 (t, $J$ = 7.1 Hz, 2H), 2.55 (t, $J$ = 7.6 Hz, 2H), 1.97 (quintet, $J$ = 7.4 Hz), 1.62–1.52 (m, 2H). [13]C NMR (100 MHz, $CDCl_3$): δ = 153.9, 134.04, 133.97, 129.56, 115.34, 54.83, 34.40, 29.35, 28.53. LC-MS m/z calc. for $C_{12}H_{15}N_3O$ [MH]$^+$; 218.1, found; 218.2, $t_R$ = 2.54 min.

## 4-(4-methoxyphenyl)butyl 4-methylbenzenesulfonate (16)

To 4-(4'-methoxyphenyl)−1-butanol (**13**) (2.00 g, 11.10 mmol, 1.0 eq) in $CHCl_3$ (30 mL) at 0°C, triethylamine (2.25 g, 22.2 mmol, 2.0 eq) and tosyl chloride (3.17 g, 16.6 mmol, 1.5 eq) were added. The reaction mixture was warmed to room temperature and left to stir for 20 hr. Sat. $NaHCO_3$(aq) was added and the organic layer was washed again with Sat. $NaHCO_3$(aq) and brine and then dried over $MgSO_4$. The solvent was removed *in vacuo* and the orange residue purified by column chromatography (1:9 EtOAc/pet. ether) to afford the title compound as a colourless oil (2.69 g, 72%). [1]H NMR (400MHz, $CDCl_3$): δ = 7.78 (d, $J$ = 8.3 Hz, 1H), 7.33 (d, $J$ = 7.9 Hz, 1H), 7.02 (d, $J$ = 8.6 Hz, 1H), 6.80 (d, $J$ = 8.6 Hz, 1H), 4.03 (t, $J$ = 6.1 Hz, 1H), 3.78 (s, 3H) 2.50 (t, $J$ = 7.2 Hz, 1H) 2.44 (s, 3H), 1.71–1.56 (m, 4H). [13]C NMR (100 MHz, $CDCl_3$): δ = 157.98, 144.79, 133.77, 133.36, 129.95, 129.36, 128.01, 113.92, 77.48, 77.36, 77.16, 76.84, 70.57, 55.40, 34.30, 28.43, 27.43, 21.77. LC-MS m/z calc. for $C_{18}H_{23}O_4S$ [MH]$^+$; 335.1, found; 335.1, $t_R$ = 3.12 min.

## 1-(4-(4-methoxyphenyl)butyl)-1H-1,2,3-triazole (17)

To 1H-1,2,3-triazole (0.225 g, 3.26 mmol, 1.0 eq) in amyl alcohol (45.0 mL) NaOH (0.130 g, 3.26 mmol, 1.0 eq) and NaI (0.489 g, 3.26 mmol, 1.0 eq) were added. The mixture was heated to reflux for 1 hr and **16** (1.200 g, 3.59 mmol, 1.1 eq) in amyl alcohol (5.0 mL) was added over 10 mins. After a further 5 hr at reflux the solvent was removed and the residue partitioned between toluene and water. The organic layer was washed with sat. $NaHCO_3$(aq) and brine and dried over $MgSO_4$. The solvent was removed *in vacuo* and the crude product purified by column chromatography (1:1 EtOAc/pet.ether). The title compound was afforded as a pale-yellow oil (0.521 g, 48%). [1]H NMR (400 MHz, $CDCl_3$): δ 7.69 (d, $J$ = 0.9 Hz, 1H), 7.50 (d, $J$ = 0.9 Hz, 1H), 7.05 (d, $J$ = 8.6 Hz, 2H), 6.82 (d, $J$ = 8.6 Hz, 2H), 4.38 (t, $J$ = 7.2 Hz, 2H), 3.78 (s, 3H), 2.59 (t, $J$ = 7.5 Hz, 2H), 2.02–1.82 (m, 2H), 1.75–1.52 (m, 2H). [13]C NMR (101 MHz, CDCl3): δ = 158.06, 133.76, 133.56, 129.37, 123.33, 114.00, 55.40, 50.23, 34.34, 29.83, 28.47.LC-MS m/z calc. for $C_{13}H_{17}N_3O$ [MH]$^+$; 231.1, found; 231.9, $t_R$ = 2.68 min.

## 1-(4-(4-hydroxyphenyl)butyl)-1H-1,2,3-triazole (18)

To a microwave vial **17** (0.46 g, 1.99 mmol, 1.0 eq) and pyridine hydrochloride (1.15 g, 9.94 mmol, 5.0 eq) were added. The mixture was heated to 180°C in the microwave for 6 min and then a further 2 min. The brown solid obtained was partitioned between EtOAc and water, and the organic layer was washed a further two times with water and then dried over $Na_2SO_4$. The solvent was removed to afford a brown oil, which was purified by column chromatography (2:98 MeOH/DCM) to afford the title compound as a pale yellow solid (0.151 g, 35%). [1]H NMR (400 MHz, DMSO-$d_6$) δ 9.11 (s, 1H), 8.10 (d, $J$ = 0.9 Hz, 1H), 7.70 (d, $J$ = 0.9 Hz, 1H), 6.94 (d, $J$ = 8.4 Hz, 2H), 6.65 (d, $J$ = 8.4 Hz, 2H), 4.38 (t, $J$ = 7.1 Hz, 2H), 2.47 (t, $J$ = 7.6 Hz, 2H), 1.79 (p, $J$ = 7.2 Hz, 2H), 1.44 (p, $J$ = 7.7 Hz, 1H). [13]C NMR (101 MHz, DMSO-$d_6$) δ 155.29, 133.11,

131.71, 129.02, 124.56, 115.00, 48.91, 33.51, 29.33, 28.07. LC-MS m/z calc. for $C_{12}H_{15}N_3O$ [MH]$^+$; 218.1, found; 217.9, $t_R$ = 2.39 min.

## 4-(4-methoxyphenyl)-1-butyl bromide (19)

To 4-(4′-methoxyphenyl)−1-butanol (13) (0.500 g, 2.78 mmol, 1.0 eq) and carbon tetrabromide (1.021 g, 3.05 mmol, 1.1 eq) in DCM (7.0 mL), triphenylphosphine (0.582 g, 2.22 mmol, 0.8 eq) was added. The mixture was stirred at room temperature overnight and the solvent was removed *in vacuo*. The crude orange residue was purified by column chromatography on silica (100% pet. ether to 1:4 EtOAc/pet. ether) to afford a colourless oil (0.335 g, 50%). $^1$H NMR (400MHz, CDCl$_3$): δ = 7.10 (d, J = 8.8 Hz, 2H), 6.84 (d, J = 8.6 Hz, 2H), 3.79 (s, 3H), 3.42 (t, J = 6.7 Hz, 2H), 2.59 (t, J = 7.5 Hz, 2H), 1.95–1.83 (m, 2H), 1.82–1.68 (m, 2H). $^{13}$C NMR (101 MHz, CDCl$_3$): δ 157.98, 134.00, 129.39, 113.94, 55.41, 34.20, 33.84, 32.36, 30.22. LC-MS m/z calc. for $C_{11}H_{15}BrO$ [MH]$^+$; 242.0, found; did not ionise, $t_R$ = 3.23 min.

## 1-(4-(4-methoxyphenyl)butyl)-1*H*-imidazole (20)

To 1I-imidazole (0.134 g, 1.97 mmol, 1.0 eq) in anhydrous DMF (2.0 mL) under nitrogen and with ice cooling, sodium hydride (0.079 g, 1.97 mmol, 1.0 eq) was added. The mixture was stirred for 30 min and 19 (0.480 g, 1.97 mmol, 1.0 eq) in DMF (2.0 mL) was added with ice cooling. The reaction mixture was stirred for a further 18 hr and then quenched with water. The solvent was removed under vacuum and the residue purified by column chromatography on silica (5:95 MeOH/DCM) to afford the title product as a pale-yellow oil (0.310 g, 68%). $^1$H NMR (400 MHz, CDCl$_3$): δ 7.43 (s, 1H), 7.07–7.02 (m, 3H), 6.87 (t, J = 1.3 Hz, 1H), 6.82 (d, J = 8.6 Hz, 2H), 3.91 (t, J = 7.1 Hz, 2H), 3.78 (s, 3H), 2.57 (t, J = 7.5 Hz, 2H), 1.85–1.71 (m, 2H), 1.63–1.53 (m, 2H). $^{13}$C NMR (101 MHz, CDCl$_3$): δ 158.03, 137.16, 133.66, 129.54, 129.35, 118.88, 113.98, 55.40, 47.03, 34.48, 30.65, 28.60. LC-MS m/z calc. for $C_{14}H_{18}N_2O$ [MH]$^+$; 230.1, found; 230.8, $t_R$ = 1.99 min.

## 1-(4-(4-methoxyphenyl)butyl)-1*H*-pyrrole (21)

To pyrrole (0.135 g, 2.02 mmol, 1.0 eq) in anhydrous DMF (2.0 mL) under nitrogen and with ice cooling, sodium hydride (0.081 g, 2.02 mmol, 1.0 eq) was added. The mixture was stirred for 30 min and 19 (0.490 g, 1.97 mmol, 1.0 eq) in DMF (2.0 mL) was added with ice cooling. The reaction mixture was stirred for a further 18 hr and then quenched with water. The solvent was removed under vacuum and the residue was purified by column chromatography on silica (5:95 MeOH/DCM) to afford the title product as a colourless oil (0.189 g, 41%). $^1$H NMR (400 MHz, CDCl$_3$): δ 7.07 (d, J = 8.7 Hz, 2H), 6.83 (d, J = 8.6 Hz, 2H), 6.64 (t, J = 2.1 Hz, 2H), 6.14 (t, J = 2.1 Hz, 2H), 3.88 (t, J = 7.1 Hz, 2H), 3.80 (s, 3H) 2.57 (t, J = 7.6 Hz, 2H), 1.89–1.74 (m, 2H), 1.67–1.56 (m, 2H). $^{13}$C NMR (101 MHz, CDCl$_3$): δ 157.93, 134.13, 129.39, 120.59, 113.90, 107.97, 55.40, 49.62, 34.64, 31.19, 28.86. LC-MS m/z calc. for $C_{15}H_{19}NO$ [MH]$^+$; 229.1, found; 230.1, $t_R$ = 3.15 min.

## 1-(4-(4-hydroxyphenyl)butyl)-1*H*-imidazole (22)

To a microwave vial 20 (0.21 g, 0.89 mmol, 1.0 eq) and pyridine hydrochloride (0.51 g, 4.45 mmol, 5.0 eq) were added. The mixture was heated to 180˚C in the microwave for 6 min. The brown solid obtained was partitioned between EtOAc and water. The organic layer was washed twice with water and then dried over Na$_2$SO$_4$. As some product remained in the aqueous layer it was basified to pH8 and re-extracted with EtOAc. The solvent was removed to afford a brown solid, which was purified by column chromatography (5:95 MeOH/DCM) to afford the title compound as a white solid (98.6 mg, 51%). $^1$H NMR (400 MHz, CDCl$_3$): δ = 7.47 (s, 1H), 7.07 (s, 1H), 6.95 (d, J = 8.4 Hz, 2H), 6.88 (s, 1H), 6.79 (d, J = 8.5 Hz, 2H), 3.92 (t, J = 7.0 Hz, 2H), 2.54 (t, J = 7.4 Hz, 2H), 1.79 (p, J = 7.1 Hz, 2H), 1.55 (p, J = 7.6 Hz, 2H). $^{13}$C

NMR (101 MHz, CDCl$_3$): δ = 155.62, 136.80, 132.34, 129.37, 128.74, 119.12, 115.74, 77.48, 77.36, 77.16, 76.84, 47.26, 34.49, 30.41, 28.55. LC-MS m/z calc. for C$_{13}$H$_{17}$N$_2$O [MH]$^+$; 217.2, found, 217.3 $t_R$ = 1.02 min.

## 1-(4-(4-hydroxyphenyl)butyl)-1*H*-pyrrole (23)

To a microwave vial **21** (0.14 g, 0.61 mmol, 1.0 eq) and pyridine hydrochloride (0.35 g, 3.05 mmol, 5.0 eq) were added. The mixture was heated to 180°C in the microwave for 6 min. The brown solid obtained was partitioned between EtOAc and water, and the organic layer was washed twice with water and then dried over Na$_2$SO$_4$. The solvent was removed to afford a brown oil, which was purified by column chromatography (1:9 to 2:8 EtOAc/pet. ether) to afford the title compound as a colourless oil (81.9 mg, 63%). $^1$H NMR (400MHz, CDCl$_3$): δ = 7.01 (d, J = 8.7 Hz, 1H), 6.74 (d, J = 8.5 Hz, 1H), 6.64 (t, J = 2.1 Hz, 2H), 6.14 (t, J = 2.1 Hz, 2H), 3.87 (t, J = 7.1 Hz, 2H), 2.55 (t, J = 7.6 Hz, 2H), 1.84–1.73 (m, 2H), 1.61–1.53 (m, 2H). $^{13}$C NMR (100 MHz, CDCl$_3$): δ = 153.76, 134.27, 129.57, 120.60, 115.29, 107.98, 77.47, 77.36, 77.16, 76.84, 49.61, 34.64, 31.16, 28.84. m/z calc. for C$_{14}$H$_{18}$NO [MH]$^+$; 216.2, found; 216.1, $t_R$ = 2.89.

## Synthesis of propenamides (28-31)

### General method A with (E)-3-phenyl-2-propenamide (31) as an example

To *trans*-cinnamic acid (**27**) (1.0 g, 6.8 mmol, 1.0 eq) in THF (10 mL) at 0°C under nitrogen, DMF (one drop) and oxalyl chloride (1.0 g, 8.1 mmol, 1.2 eq) were added. The mixture was stirred at room temperature for 2 hr and further oxalyl chloride (0.5 eq) was added. The solvent was removed *in vacuo* and the resulting oil dissolved in EtOAc (5 mL) and added dropwise to a stirred mixture of 35% NH$_4$OH(aq) (5.0 mL) and EtOAc (2.0 mL) at 0°C. The resulting white needle-like crystals were recovered by vacuum filtration and washed with water and petroleum ether to afford the title compound (0.891 g, 90%). $^1$H NMR (400MHz, DMSO-*d$_6$*): δ = 7.59–7.52 (m, 3H), 7.47–7.33 (m, 4H), 7.13 (broad s, 1H), 6.62 (d, J = 15.9 Hz, 1H). $^{13}$C NMR (100 MHz, DMSO-*d$_6$*): δ = 166.68, 139.15, 134.88, 129.42, 128.90, 127.52, 122.34. LC-MS m/z calc. for C$_9$H$_9$NO [MH]$^+$; 148.1, found; 148.1, $t_R$ = 2.18 min.

### (*E*)-3-(4-methylphenyl)-2-propenamide (28)

Compound **28** was prepared according to the procedure described in general method A using (*E*)−3-(p-tolyl)acrylic acid (**24**) as starting material. White solid (0.831 g, 84%). $^1$H NMR (400MHz, DMSO-*d$_6$*): δ = 7.51 (broad s, 1H), 7.45 (d, J = 8.1 Hz, 2H), 7.38 (d, J = 15.9 Hz, 1H), 7.21 (d, J = 7.9 Hz, 2H), 7.08 (broad s, 1H), 6.56 (d, J = 15.9 Hz, 1H), 2.31 (s, 3H). $^{13}$C NMR (100 MHz, DMSO-*d$_6$*): δ = 166.83, 139.15, 139.09, 132.13, 129.49, 127.79, 121.29, 20.91. LC-MS m/z calc. for C$_{10}$H$_{12}$NO [MH]$^+$; 162.1, found; 162.2, $t_R$ = 2.35 min.

### (*E*)-3-(4-methoxyphenyl) propenamide (29)

Compound **29** was prepared according to the procedure described in general method A using (*E*)−3-(4-methoxyphenyl)acrylic acid (**25**) as starting material. White solid (0.833 g, 84%). $^1$H NMR (400MHz, DMSO-*d$_6$*): δ = 7.49 (d, J = 8.8 Hz, 2H), 7.46 (broad s, 1H), 7.37 (d, J = 15.8 Hz, 1H), 7.01 (broad s, 1H), 6.96 (d, J = 8.8 Hz, 2H), 6.47 (d, J = 15.9 Hz 1H), 3.78 (s, 3H). $^{13}$C NMR (100 MHz, DMSO-*d$_6$*): δ = 167.00, 160.31, 138.88, 129.09, 127.43, 119.81, 114.36, 55.23. LC-MS m/z calc. for C$_{10}$H$_{11}$NO$_2$ [MH]$^+$; 178.1, found; 178.2, $t_R$ = 2.21 min.

## (E)-3-(4-trifluoromethylphenyl)-2-propenamide (30)

Compound **30** was prepared according to the procedure described in general method A using (E)−3-(4-trifluoromethylphenyl)acrylic acid (**26**) as starting material. White solid (0.850 g, 85%). $^1$H NMR (400MHz, DMSO-$d_6$): δ = 7.72 (m, 4H), 7.63 (broad s, 1H), 7.49 (d, $J$ = 15.9 Hz, 1H), 7.24 (broad s, 1H), 6.75 (d, $J$ = 15.9 Hz, 1H). $^{13}$C NMR (100 MHz, DMSO-$d_6$): δ = 166.18, 138.97, 129.20 (q, $J$ = 31.8 Hz), 128.14, 125.74 (q, $J$ = 3.8 Hz), 125.18, 122.73 (q, $J$ = 272.1 Hz). LC-MS m/z calc. for $C_{10}H_9F_3NO$ [MH]$^+$; 216.0, found; 216.2, $t_R$ = 2.62 min.

## Synthesis of oxazole intermediates (32-35)

### General method B with 4-chloromethyl-2-[(E)-2-phenylethenyl]-oxazole (35) as an example

To **31** (0.500g, 3.39 mmol, 1.0 eq) in toluene (5.0 mL), 1,3-dichloroacetone (0.431 g 3.39 mmol, 1.0 eq) was added. The mixture was stirred at reflux for 4.5 hr and after cooling was poured onto sat. $K_2CO_3$(aq). The aqueous layer was extracted with EtOAc and the combined organic layers were washed with water, brine, and then dried over $Na_2SO_4$. The crude product was then purified by column chromatography (1:9 EtOAc/pet. ether) to afford the title product as a white crystalline solid (0.323 g, 43%). $^1$H NMR (400MHz, CDCl$_3$): δ = 7.62 (s, 1H), 7.58–7.50 (m, 3H), 7.43–7.31 (m, 3H), 6.92 (d, $J$ = 16.37, 1H), 4.54 (d, $J$ = 0.8 Hz, 2H). $^{13}$C NMR (100 MHz, CDCl$_3$): δ = 162.49, 139.14, 137.43, 136.10, 135.61, 129.76, 129.26, 127.62, 113.79, 37.34. LC-MS m/z calc. for $C_{12}H_{11}^{35}ClNO$ [MH]$^+$; 220.1, found; 220.1, $t_R$ = 2.88 min.

### 4-chloromethyl-2-[(E)−2-(4-methylphenyl)ethenyl]-1,3-oxazole (32)

Compound **32** was prepared according to the procedure described in general method B using **28** as starting material. A further 0.5 eq of 1,3-dichloroacetone was added after 24 hr. Column chromatography eluent; 1:9 EtOAc/pet. ether. The title compound was obtained as a white solid (0.444 g, 61%). $^1$H NMR (400MHz, CDCl$_3$): δ = 7.61 (s, 2H), 7.51 (d, $J$ = 16.4 Hz, 1H), 7.42 (d, $J$ = 8.1 Hz, 2H), 7.19 (d, $J$ = 8.0 Hz, 2H), 6.87 (d, $J$ = 16.4 Hz, 1H), 4.53 (d, $J$ = 0.8 Hz, 2H), 2.37 (s, 3H). $^{13}$C NMR (100 MHz, CDCl$_3$): δ = 162.51, 139.83, 138.84, 137.21, 135.74, 132.67, 129.77, 127.34, 112.56, 37.18, 21.53. LC-MS m/z calc. for $C_{13}H_{13}^{35}ClNO$ [MH]$^+$; 234.1 found; 234.4, $t_R$ = 3.09 min.

### 4-chloromethyl-2-[(E)-2-(4-methoxyphenyl)ethenyl]-1,3-oxazole (33)

Compound **33** was prepared according to the procedure described in general method B using **29** as starting material. Column chromatography eluent; 1:9 EtOAc/pet. ether. The title compound was obtained as a pale yellow solid (0.238 g, 34%). $^1$H NMR (400 MHz, CDCl$_3$) δ 7.60 (s, 1H), 7.51 (s, 1H), 7.49 (d, $J$ = 16.3 Hz 7.47 (d, $J$ = 8.5 Hz, 1H), 6.92 (d, $J$ = 8.8 Hz, 2H), 6.78 (d, $J$ = 16.4 Hz, 1H), 4.53 (d, $J$ = 1.0 Hz, 2H), 3.84 (s, 3H). $^{13}$C NMR (100 MHz, CDCl$_3$): δ = 162.56, 161.03, 138.95, 137.02, 135.77, 129.07, 128.36, 114.69, 111.52, 55.69, 37.39. LC-MS m/z calc. for $C_{13}H_{13}^{35}ClNO_2$ [MH]$^+$; 250.0, found; 250.2, $t_R$ = 2.85 min.

### 4-chloromethyl-2-[(E)-2-(4-trifluoromethylphenyl)ethenyl]-1,3-oxazole (34)

Compound **34** was prepared according to the procedure described in general method B using **30** as starting material. Column chromatography eluent; 1:9 EtOAc/pet. ether. The title compound was obtained as a white solid (0.359 g, 54%). $^1$H NMR (400MHz, CDCl$_3$): δ = 7.68–7.59 (m, 5H), 7.56 (d, $J$ = 7.6 Hz, 1H), 6.99 (d, $J$ = 16.4 Hz, 1H), 4.54 (d, $J$ = 0.8 Hz, 2H). $^{13}$C NMR (100 MHz, CDCl$_3$): δ = 161.63, 139.25, 138.79, 136.33, 135.39, 131.28, 130.95, 127.51, 126.04 (q, $J$ = 3.9 Hz), 124.07 (q, $J$ = 272.24), 115.94, 37.00, LC-MS: m/z calc. for $C_{13}H_{10}^{35}ClF_3NO$ [MH]$^+$; 288.04, found; 287.9, $t_R$ = 3.16 min.

### 4-(4'-methylphenoxymethyl)-2-[(E)-2-(4-trifluoromethylphenyl) ethenyl]-1,3-oxazole (5)

To a solution of **34** (0.12 g, 0.42 mmol, 1.0 eq) in DMF (5.0 mL), sodium hydride (60% in mineral oil, 17.0 mg, 0.42 mmol, 1.0 eq) was added under $N_2$ with ice cooling. The mixture was stirred at room temperature for 30 min and *p*-cresol (45.0 mg, 0.42 mmol, 1.0 eq) was added. The mixture was stirred at room temperature for 23 hr and water (20 mL) was then added. A precipitate formed which was isolated by vacuum filtration, washed with water and petroleum ether and was then purified by column chromatography (1:9 EtOAc/pet. ether). The solvent was removed in vacuo to afford the title compound as a white crystalline solid. $^1$H NMR (400MHz, CDCl$_3$): 7.67 (s, 1H), 7.67–7.59 (m, 4H), 7.54 (d, *J* = 16.5 Hz, 1H), 7.10 (d, *J* = 8.3 Hz, 2H), 7.01 (d, *J* = 16.4 Hz, 1H), 6.89 (d, *J* = 8.6 Hz, 2H), 5.02 (d, *J* = 0.98, 2H), 2.29 (s, 3H) $^{13}$C NMR (100 MHz, CDCl$_3$): δ = 161.38, 156.33, 139.03, 138.94, 136.57, 134.92, 130.93 (q, *J* = 29.1 Hz), 130.12, 127.47, 126.02 (q, *J* = 3.9 Hz), 121.41 (q, *J* = 270.9 Hz), 116.18, 114.81, 62.83, 20.64. HRMS: (m/z) calc for $C_{20}H_{16}F_3NO_2$ [MH]$^+$; 360.1206 found; 360.1213. LC-MS m/z calc. for $C_{20}H_{16}F_3NO_2$ [MH]$^+$; 360.1, found; 360.2, $t_R$ = 3.35 min. Analytical RP-HPLC $t_R$ = 22.35, 95% purity.

## Synthesis of triazole, pyrrole and imidazole final compounds (1–4 and 6–8)

### *General method C with (E)-4-((4-(4-(2H-1,2,3-triazol-2-yl)butyl) phenoxy)methyl)-2-(4-(trifluoromethyl)styryl)oxazole (6) as an example*

To a solution of **15** (30.0 mg, 0.14 mmol, 1.0 eq) in DMF (2.0 mL), sodium hydride (60% in mineral oil, (55.0 mg, 0.14 mmol, 1.0 eq) was added with ice cooling under $N_2$. The mixture was stirred at room temperature for 30 min and **34** (39.0 mg, 0.14 mmol, 1.0 eq) was added with ice cooling. The mixture was stirred at room temperature for 19 hr and water was then added. A precipitate formed which was isolated by vacuum filtration and was then purified by column chromatography (2:8 EtOAc/pet. ether). The title compound was obtained as a white solid (20.0 mg, 31%). $^1$H NMR (400MHz, CDCl$_3$): δ = 7.67 (s, 1H), 7.66–7.59 (m, 4H), 7.54 (d, *J* = 16.3 Hz, 1H) 7.58 (s, 1H), 7.08 (d, *J* = 8.6 Hz, 2H), 7.01 (d, *J* = 16.4 Hz, 1H), 6.91 (d, *J* = 8.6 Hz, 2H), 5.02 (s, 1H), 5.01 (s, 1H), 4.46 (t, *J* = 7.0 Hz, 2H), 2.59 (t, *J* = 7.6 Hz, 2H), 1.99 (quintet, *J* = 7.4 Hz, 2H), 1.65–1.54 (m, 2H). $^{13}$C NMR (100 MHz, CDCl$_3$): δ = 161.25, 156.56, 138.86, 138.79, 136.44, 134.79, 134.63, 133.89, 130.88 (q, *J* = 31.1 Hz), 129.34, 127.34, 125.88 (q, *J* = 3.8 Hz), 123.96 (q, *J* = 271.6 Hz), 116.05, 114.73, 62.67, 54.66, 34.31, 29.24, 28.36. HRMS: (m/z) calc. for $C_{25}H_{23}N_4O_2$ [MH]$^+$; 469.1846 found; 469.1849. LC-MS m/z calc. for $C_{25}H_{23}N_4O_2$ [MH]$^+$; 469.2, found; 469.2, $t_R$ = 3.21 min. Analytical RP-HPLC: $t_R$ = 22.26, Purity = 99%.

### (E)-4-((4-(4-(1H-1,2,3-triazol-1-yl)butyl)phenoxy)methyl)-2-((4-trifluoromethyl)styryl)oxa-zole (1)

Compound one was prepared according to the procedure described in general method C using compounds **18** and **34** as starting materials. Column chromatography eluent (1:1 EtOAc/pet. ether) the title compound was obtained as a white solid (39.0 mg, 60%). $^1$H NMR (400 MHz, DMSO-$d_6$):) δ 8.24 (s, 1H), 8.11 (d, *J* = 0.9 Hz, 1H), 7.95 (d, *J* = 8.1 Hz, 2H), 7.76 (d, *J* = 8.2 Hz, 2H), 7.70 (d, *J* = 0.9 Hz, 1H), 7.61 (d, *J* = 16.4 Hz, 1H), 7.34 (d, *J* = 16.5 Hz, 1H), 7.09 (d, *J* = 8.6 Hz, 2H), 6.94 (d, *J* = 8.6 Hz, 2H), 4.98 (d, *J* = 0.9 Hz, 2H), 4.39 (t, *J* = 7.1 Hz, 2H), 2.54 (d, *J* = 7.6 Hz, 2H), 1.81 (m, 2H), 1.60–1.39 (m, 2H). $^{13}$C NMR (101 MHz, DMSO-$d_6$):) δ = 160.55, 156.18, 139.19, 138.11, 138.08, 134.30, 134.07, 133.12, 129.18, 128.05, 125.63 (q, *J* = 3.4 Hz), 124.58, 124.11 (q, *J* = 271.8), 116.42, 114.55, 61.49, 48.88, 33.44, 29.32, 27.96. LC-MS m/z calc. for $C_{25}H_{23}N_4O_2$ [MH]$^+$; 469.2, found; 469.1, $t_R$ = 3.09 min. HRMS: (m/z) calc. for $C_{25}H_{23}N_4O_2$ [MH]$^+$; 469.1846 found; 469.1853. Analytical RP-HPLC: $t_R$ = 20.19 min, purity = 99%.

### (*E*)-4-((4-(4-(1H-1,2,3-triazol-1-yl)butyl)phenoxy)methyl)-2-(styryl)oxazole (2)

Compound two was prepared according to the procedure described in general method C using compounds **18** and **35** as starting materials. Column chromatography eluent; 7:3 EtOAc/pet. ether. The title compound was obtained as an off-white solid (0.05 g, 68%). $^1$H NMR (400 MHz, CDCl$_3$) δ 7.71 (s, 1H), 7.65 (s, 1H), 7.61–7.46 (m, 4H), 7.44–7.31 (m, 3H), 7.07 (d, *J* = 8.3 Hz, 2H), 6.95 (d, *J* = 16.5 Hz, 1H), 6.92 (d, *J* = 8.5 Hz, 2H), 5.02 (s, 2H), 4.39 (t, *J* = 7.1 Hz, 2H), 2.60 (t, *J* = 7.5 Hz, 2H), 1.94 (m, 2H), 1.63 (m, 2H). $^{13}$C NMR (101 MHz, CDCl$_3$) δ = 162.08, 156.79, 138.14, 137.63, 136.27, 135.34, 134.29, 133.59, 129.68, 129.49, 129.07, 127.51, 123.61, 114.99, 113.28, 62.65, 50.48, 34.39, 29.84, 28.43. LC-MS *m/z* calc. for C$_{24}$H$_{24}$N$_4$O$_2$ [MH]$^+$; 401.2, found; 401.1, $t_R$ = 3.02 min. HRMS: (m/z) calc. for C$_{24}$H$_{24}$N$_4$O$_2$ [MH]$^+$; 401.1972 found; 401.1976. Analytical RP-HPLC $t_R$ = 18.99 min, purity = 98%.

### (*E*)−4-((4-(4-(1H-1,2,3-triazol-1-yl)butyl)phenoxy)methyl)-2-((4-methyl)styryl)oxazole (3)

Compound three was prepared according to the procedure described in general method C using **18** and **32** as starting materials. Column chromatography eluent; 8:2 EtOAc/pet. ether. The title compound was obtained as a white solid (38.0 mg, 50%). $^1$H NMR (400 MHz, CDCl$_3$): δ 7.70 (s, 1H), 7.64 (s, 1H), 7.53 (d, *J* = 16.4 Hz, 1H), 7.50 (s, 1H), 7.43 (d, *J* = 8.0 Hz, 2H), 7.20 (d, *J* = 7.9 Hz, 2H), 7.07 (d, *J* = 8.6 Hz, 2H), 6.91 (d, *J* = 8.6 Hz, 1H), 6.89 (d, *J* = 16.4 Hz, 1H), 5.01 (d, *J* = 1.1 Hz, 2H), 4.39 (t, *J* = 7.2 Hz, 2H), 2.60 (t, *J* = 7.5 Hz, 2H), 2.38 (s, 3H), 2.05–1.87 (m, 2H), 1.75–1.52 (m, 2H). $^{13}$C NMR (101 MHz, CDCl$_3$): δ = 162.30, 156.79, 140.05, 137.97, 137.77, 136.11, 134.30, 133.66, 132.59, 129.80, 129.47, 127.50, 123.45, 114.98, 112.17, 62.61, 50.35, 34.38, 29.83, 28.42, 21.58. LC-MS *m/z* calc. for C$_{25}$H$_{26}$N$_4$O$_2$ [MH]$^+$; 415.2, found; 415.1, $t_R$ = 3.08 min. HRMS: (m/z) calc. for C$_{25}$H$_{26}$N$_4$O$_2$ [MH]$^+$; 415.2129 found; 415.2132. Analytical RP-HPLC $t_R$ = 19.68 min, purity = 97%.

### (*E*)-4-((4-(4-(1 H-1,2,3-triazol-1-yl)butyl)phenoxy)methyl)-2-((4-methoxy)styryl)oxazole (4)

Compound four was prepared according to the procedure described in general method C using **18** and **33** as starting materials. Column chromatography eluent; (8:2 EtOAc/pet. ether) the title compound was obtained as a white solid (48.4 mg, 81%). $^1$H NMR (400 MHz, CDCl3): δ 7.69 (d, *J* = 1.0 Hz, 1H), 7.62 (s, 1H), 7.50 (d, *J* = 16.6 Hz, 1H), 7.49 (d, *J* = 1.0 Hz, 1H), 7.48 (d, *J* = 8.6 Hz, 2H), 7.06 (d, *J* = 8.6 Hz, 2H), 6.92 (d, *J* = 8.8 Hz, 2H), 6.91 (d, *J* = 8.6 Hz, 2H), 6.80 (d, *J* = 16.4 Hz, 1H), 5.00 (d, *J* = 1.1 Hz, 2H), 4.39 (t, *J* = 7.2 Hz, 2H), 3.84 (s, 3H), 2.60 (t, *J* = 7.5 Hz, 2H), 1.97–1.90 (m, 2H), 1.67–1.56 (m, 2H). 13C NMR (101 MHz, CDCl3): δ 162.47, 160.97, 156.78, 137.87, 137.36, 135.93, 134.30, 133.80, 129.46, 129.04, 128.09, 123.11 (q, *J* = 271.8 Hz) 123.35, 114.95, 114.54, 110.90, 62.61, 55.54, 50.25, 34.39, 29.85, 28.43. LC-MS m/z calc. for C$_{25}$H$_{23}$N$_4$O$_2$ [MH]$^+$; 431.2, found; 431.0, $t_R$ = 3.00 min. HRMS: (m/z) calc. for C$_{25}$H$_{23}$N$_4$O$_2$ [MH]$^+$; 431.2078 found; 431.2079. Analytical RP-HPLC: $t_R$ = 18.83 min, purity = 99%.

### (*E*)-4-((4-(4-(1*H*-imidazol-1-yl)butyl)phenoxy)methyl)-2-(4-(trifluoromethyl)styryl)oxazole (7)

Compound seven was prepared according to the procedure described in general method C using **22** and **34** as starting materials. Column chromatography eluent; 3:7 EtOAc/pet ether. The title compound was obtained as a white solid (61.8 mg, 95%). $^1$H NMR (400MHz, CDCl$_3$): δ = 7.68 (br s, 1H), 7.67–7.59 (m, 4H), 7.55 (d, *J* = 16.5 Hz, 1H), 7.44 (br s, 1H)7.06 (d, *J* = 8.6 Hz, 2H), 7.04 (br s, 1H) 7.01 (d, *J* = 16.4 Hz, 1H), 6.92 (d, *J* = 8.7 Hz, 3H), 6.87 (br s, 1H), 5.02 (d, *J* = 1.0 Hz, 2H), 3.92 (t, *J* = 7.1 Hz, 2H), 2.58 (t, *J* = 7.5 Hz, 2H), 1.79 (m, 2H), 1.59 (m, 2H). $^{13}$C NMR (100 MHz, CDCl$_3$): δ = 161.40, 156.77, 138.88, 137.16, 136.59, 134.97, 134.43,

129.52, 129.44, 127.46, 126.00 (q, $J = 3.6$ Hz), 118.89, 116.13, 114.92, 77.47, 77.36, 77.15, 76.84, 62.76, 47.04, 34.51, 30.64, 28.54. HRMS: (m/z) calc. for $C_{26}H_{24}F_3N_3O_2$ [MH][+]; 468.1893 found; 468.1914. LC-MS m/z calc. for $C_{26}H_{24}F_3N_3O_2$ [MH][+]; 468.2, found; 468.3, $t_R$ = 2.50 min. Analytical RP-HPLC: $t_R$ = 17.64, purity = 97%.

### (E)-4-((4-(4-(1H-pyrrol-1-yl)butyl)phenoxy)methyl)-2-(4-(trifluoromethyl)styryl)oxazole (8)

Compound eight was prepared according to the procedure described in general method C using **23** and **34** as starting materials. Column chromatography eluent; 1:9 to 2:8 EtOAc/pet ether. The title compound was obtained as an off-white solid (55.1 mg, 85%). [1]H NMR (400MHz, CDCl₃): δ = 7.68 (s, 1H), 7.64 (m, 4H), 7.55 (d, $J = 16.4$ Hz, 1H), 7.08 (d, $J = 8.6$ Hz, 2H), 7.02 (d, $J = 16.4$ Hz, 1H), 6.91 (d, $J = 8.6$ Hz, 1H), 6.63 (t, $J = 2.1$ Hz, 2H), 6.13 (t, $J = 2.1$ Hz, 2H), 5.02 (d, $J = 1.1$ Hz, 2H), 3.87 (t, $J = 7.1$ Hz, 2H), 2.57 (t, $J = 7.6$ Hz, 2H), 1.88–1.71 (m, 2H), 1.65–1.57 (m, 1H). [13]C NMR (100 MHz, CDCl₃): δ = 161.39, 156.67, 138.97, 136.58, 134.95, 134.91, 129.49, 127.47, 126.02 (q, $J = 3.6$ Hz), 120.60, 116.17, 114.85, 108.00, 77.48, 77.36, 77.16, 76.84, 62.80, 49.62, 34.68, 31.19, 28.80. HRMS: (m/z) calc. for $C_{27}H_{25}F_3N_2O_2$ [MH][+]; 467.1941 found; 467.1944. LC-MS m/z calc. for $C_{27}H_{25}F_3N_3O_2$ [MH][+]; 467.2, found; 467.5, $t_R$ = 3.48 min. Analytical RP-HPLC: $t_R$ = 23.23, purity = 97%.

## Carboxyamidotriazole (CAI) Analogues

**Scheme 3.** *Reagents & Conditions:* (a) TBDMSCl, imidazole, DMAP, DMF; (b) (i) nBuLi, THF, -78 °C, 4-chlorobenzoyl chloride; (c) TBAF, THF; (d) PBr₃, THF; (e) Tosyl chloride, DIPEA, CHCl₃; (f) 5-amino-1H-imidazole-4-carboxamide, K₂CO₃, MeCN; (g) 3-aminopyrazole-carboxamide, NaH, DMF; (h) NaN₃, EtOH (i) Cyanoacetamide, EtOH, 30% NaOMe in MeOH.

**Appendix 1—figure 2.** Carboxyamidotriazole analogues.

Scheme 3. Reagents and Conditions: (a) TBDMSCl, imidazole, DMAP, DMF; (b) (i) *n*BuLi, THF, −78 °C, 4-chlorobenzoyl chloride; (c) TBAF, THF; (d) PBr₃, THF; (e) Tosyl chloride, DIPEA, CHCl₃; (f) 5-amino-1H-imidazole-4-carboxamide, K₂CO₃, MeCN; (g) 3-aminopyrazole-carboxamide, NaH, DMF; (h) NaN₃, EtOH (i) Cyanoacetamide, EtOH, 30% NaOMe in MeOH.

### tert-butyl((3,5-dichlorobenzyl)oxy)dimethylsilane (37)

To a mixture of 3,5-dibromobenzyl alcohol (**36**) (3.00 g, 16.9 mmol) and imidazole (2.77 g, 40.7 mmol) in DMF (30 mL), was added *tert*-butyldimethylsilyl chloride (3.06 g, 20.3 mmol). After stirring for 1 day at room temperature, the mixture was diluted with ether, washed twice with water and then dried over MgSO₄. Concentration of the organic phase afforded a clear colourless oil (4.94 g, 100%) [1]H NMR (400 MHz, Chloroform-d) δ 7.25–7.22 (m, 1H), 7.21–7.18 (m, 2H), 4.68 (s, 2H), 0.95 (s, 9H), 0.11 (s, 6H). [13]C NMR (101 MHz, CDCl₃) δ 145.12, 134.93, 127.10, 124.44, 63.88, 26.04, 18.53,–5.18.

### (4-(((*tert*-butyldimethylsilyl)oxy)methyl)-2,6-dichlorophenyl)(4-chlorophenyl)methanone (38)

A mixture of **37** (1.00 g, 3.4 mmol) in anhydrous THF (10 mL) was cooled to −72°C in a hexane/dry ice bath. *n*-Butyl lithium (2.3M, 1.5 mL, 3.8 mmol) was added dropwise keeping the temperature ≤−60°C. The mixture was stirred at ≤60°C for 30 min and was then cooled to −72°C. Chlorobenzoyl chloride (0.52 mL, 4.1 mmol) in THF (2 mL) was added dropwise keeping the temperature below −60°C. The mixture was stirred at −60°C for 3 hr and was then quenched with 2M HCl$_{(aq)}$ (2 mL) and allowed to warm to room temperature. The THF was removed under vacuum and the residue was partitioned between EtOAc and water. The organic layer was washed with sat. NaHCO$_3$, brine and then dried over MgSO$_4$. The solvent was removed to afford a pale-yellow oil containing small crystals. The crude product was loaded onto isolute and purified by column chromatography (100% pet. ether to 20% DCM). The solvent was removed to afford a pale yellow oil (0.500 g, 34%). $^1$H NMR (400 MHz, Chloroform-*d*) δ 7.76 (d, *J* = 8.6 Hz, 2H), 7.45 (d, *J* = 8.6 Hz, 2H), 7.34 (s, 2H), 4.75 (t, *J* = 0.9 Hz, 2H), 0.97 (s, 9H), 0.14 (s, 6H). $^{13}$C NMR (101 MHz, CDCl$_3$) δ 191.68, 145.76, 140.96, 135.58, 134.16, 131.86, 131.10, 129.48, 125.52, 63.60, 26.04, 18.55,–5.17. LC-MS *m/z* calc. for C$_{20}$H$_{24}$Cl$_3$O$_2$Si [MH]$^+$; 429.1, found; 429.1, $t_R$ = 3.78 min.

### (4-Chlorophenyl)(2,6-dichloro-4-(hydroxymethyl)phenyl)methanone (39)

To a solution of **38** (0.50 g, 1.2 mmol) in THF (8 mL), TBAF (1M in THF, 2.1 mL, 2.1 mmol) was added. The mixture was stirred for 5 hr at room temperature. EtOAc was added and the solution washed twice with sat. NH$_4$Cl$_{(aq)}$ and then water. The organic phase was dried over MgSO$_4$, filtered and the solvent was removed to afford a white solid. The solid was triturated with pet. ether and collected by vacuum filtration. (0.282 g, 77%). $^1$H NMR (400 MHz, Chloroform-*d*) δ 7.76 (d, *J* = 8.6 Hz, 2H), 7.46 (d, *J* = 8.6 Hz, 2H), 7.39 (s, 2H), 4.76 (d, *J* = 4.4 Hz, 2H), 2.06 (t, *J* = 5.8 Hz, 1H). $^{13}$C NMR (101 MHz, CDCl$_3$) δ 191.60, 144.93, 141.09, 136.11, 134.03, 132.12, 131.09, 129.52, 126.18, 63.63. LC-MS *m/z* calc. for C$_{14}$H$_{10}$Cl$_3$O$_2$ [MH]$^+$; 315.0, found; 315.1, $t_R$ = 2.92 min.

### 3,5-dichloro-4-(4'-chlorobenzoyl)benzyl 4''-methylbenzenesulfonate (40)

To **39** (0.20 g, 0.63 mmol) in chloroform (3 mL) at 0°C, DIPEA (0.22 mL, 1.27 mmol) and tosyl chloride (0.18 g, 0.95 mmol) were added. The mixture was allowed to warm to room temperature and was stirred for 24 hr. 0.5 equivalents (0.06 g, 0.32 mmol) of tosyl chloride were added and the mixture was stirred for a further 24 hr. Sat. NaHCO$_3$ and chloroform were added, and the organic layer was washed again with sat. NaHCO$_3$ and brine and then dried over MgSO$_4$. Concentration under vacuum afforded a brown oil which was purified by column chromatography (1:9 EtOAc/pet. ether). The solvent was removed to afford the title compound as a clear colourless oil which slowly crystallised to a white solid (0.232 g, 78%) $^1$H NMR (400 MHz, Chloroform-*d*) δ 7.93 (d, *J* = 8.5 Hz, 1H), 7.76 (d, *J* = 8.6 Hz, 1H), 7.47 (d, *J* = 8.6 Hz, 1H), 7.43 (s, 2H), 7.41 (d, *J* = 8.1 Hz, 1H), 4.57 (s, 2H), 2.49 (s, 2H). $^{13}$C NMR (101 MHz, CDCl$_3$) δ 191.11, 146.92, 141.85, 141.23, 141.10, 137.16, 133.81, 132.31, 131.07, 130.36, 129.58, 128.25, 127.20, 44.13, 21.97.

### (4-(Bromomethyl)-2,6-dichlorophenyl)(4'-chlorophenyl)methanone (41)

To a solution of **SJM-169-291/293** (0.23 g, 0.74 mmol) in anhydrous THF (2 mL), under a nitrogen atmosphere, phosphorus tribromide (76.3 μL, 0.81 mmol) was added. The mixture was stirred at room temperature for 2 hr and then quenched with sat. NaHCO$_{3(aq)}$ and

extracted into diethyl ether twice. The combined organic phases were washed with brine and then dried over $MgSO_4$. The solvent was removed under vacuum to afford a white solid which was purified by preparative TLC (5:95 EtOAc/pet. ether) (0.138 g, 49%). $^1H$ NMR (400 MHz, Chloroform-d) δ 7.76 (d, J = 8.6 Hz, 1H), 7.47 (d, J = 8.6 Hz, 1H), 7.42 (s, 1H), 4.43 (s, 1H). $^{13}C$ NMR (101 MHz, CDCl$_3$) δ 191.09, 141.44, 141.24, 137.16, 133.81, 132.29, 131.08, 129.58, 128.79, 30.42. LCMS; $t_R$ = 3.20 min. HRMS (ESI-TOF) m/z calc. for $C_{14}H_8$ BrCl$_3$O [M+H]$^+$; 376.8897 found; 376.8892 and 398.8716 [M+Na]$^+$.

### 5-Amino-1-(4[4-chlorobenzoyl]-3,5-dichlorobenzyl)-imidazole-4-carboxamide formate (12)

To 5-amino-1H-imidazole-4-carboxamide (0.045 g, 0.36 mmol) in DMF (2 mL) at 0°C under a nitrogen atmosphere, sodium hydride (60% in mineral oil, 14.3 mg, 0.36 mmol) was added. The mixture was allowed to warm to room temperature and was stirred for 1.25 hr. After this time **41** (0.135 g, 0.36 mmol) in DMF (1 mL) was added dropwise and the reaction mixture was stirred for 3.5 hr. The reaction was quenched with water and extracted with ethyl acetate. The organic layer was washed with water and brine and then dried over $MgSO_4$. The solvent was removed to afford a dark purple residue which was purified by column chromatography (1:9 MeOH/DCM). As a small impurity remained the lilac solid was precipitated from EtOAC/pet. ether and then further purified by semi-prep RP-HPLC (40% to 95% B over 15 mins). The title product was afforded as a white solid after lyophilisation (19.0 mg, 13%). $^1H$ NMR (400 MHz, Methanol-$d_4$) δ 8.46 (s, 1H), 7.77 (d, J = 8.6 Hz, 2H), 7.56 (d, J = 8.7 Hz, 2H), 7.34 (2 x s, 3H), 5.21 (s, 2H). $^{13}C$ NMR (101 MHz, MeOD) δ 205.02, 178.36, 142.08, 137.85, 137.41, 135.28, 133.22, 132.34, 132.12, 130.62, 128.08, 123.26, 122.31, 46.41. LCMS m/z calc. for $C_{18}H_{13}^{35}Cl_3N_4O_2$ [MH]$^+$; 423.0, found; 423.1, $t_R$ = 2.66 min. HRMS (ESI-TOF) m/z calc. for $C_{18}H_{14}^{35}Cl_3N_4O_2$ [M+H]$^+$; 423.0177 found; 423.0178 and 444.9996 [M+Na]$^+$.

### 3-Amino-1-(4[4-chlorobenzoyl]-3,5-dichlorobenzyl)-pyrazole-4-carboxamide hydroformate (10) and 5-amino-1-(3,5-dichloro-4-(4-chlorobenzoyl)benzyl)-1H-pyrazole-4-carboxamide formate (11)

To 3-aminopyrazole-4-carboxylic acid amide (0.03 g, 0.24 mmol) in DMF (2 mL) at 0°C, sodium hydride (9.5 mg, 0.24 mmol) was added. The mixture warmed to room temperature and stirred for 1 hr. **41** (0.11 g, 0.24 mmol) in DMF (1 mL) was then added dropwise at 0°C. The mixture was stirred at room temperature for 1.5 hr then heated at 90°C for 24 hr. The reaction was quenched with water and then partitioned between water and DCM. The aqueous phase was extracted with DCM and the combined organic phases were dried over $MgSO_4$ and the solvent was removed under vacuum. The resulting yellow solid was purified by column chromatography on silica (5% MeOH in DCM to 10% MeOH in DCM) to afford the title compound as a white solid (16 mg). NMR showed two products (N-1 and N-2 isomers) so these were isolated by semi-prep HPLC (35% to 47% B over 16 mins with a 1 min hold at 42% B). The products were isolated as a white solid after lyophilisation; **10** (18.5 mg, 19%) **11** (16.4 mg, 16%). $^1H$ NMR (400 MHz, DMSO-$d_6$) **10**; δ 8.53 (s, 1H), 8.05 (s, 1H), 7.77 (d, J = 8.5 Hz, 2H), 7.67 (d, J = 8.5 Hz, 2H), 7.50 (s, 2H), 5.46 (s, 1H), 5.20 (s, 2H). $^1H$ NMR (400 MHz, DMSO-$d_6$) **11**; δ 8.53 (s, 1H), 7.80–7.71 (m, 3H), 7.67 (d, J = 8.7 Hz, 2H), 7.37 (s, 2H), 6.44 (s, 2H), 5.25 (s, 2H). $^{13}C$ NMR (101 MHz, MeOD); **10**; δ = 192.42, 170.32, 158.79, 142.55, 142.12, 135.31, 133.19, 133.00, 132.11, 130.61, 128.62, 127.43, 102.62, 54.88. $^{13}C$ NMR (101 MHz, MeOD); **11**; δ = 192.46, 170.32, 151.84, 146.83, 142.48, 142.10, 139.84, 137.57, 135.33, 133.00, 132.11, 130.60, 128.13, 98.26, 63.18. LCMS m/z calc. for $C_{18}H_{13}Cl_3N_4O_2$ [MH]$^+$; 423.0, found; 423.1, $t_R$ = 3.18 min. HRMS (ESI-TOF) m/z calc. for $C_{18}H_{14}Cl_3N_4O_2$ [M+H]$^+$; 423.0177 found; 423.0179 and 444.9996 [M+Na]$^+$. Analytical HPLC; (**10**) $t_R$ = 16.53 mins, purity = 100%, (**11**) $t_R$ = 16.83 mins, purity = 99%.

### 3,5-dichloro-4-(4-chlorobenzoyl) benzyl azide (42)

To **41** (50.0 mg, 0.13 mmol) in ethanol (1 mL), sodium azide (17.2 mg, 0.26 mmol) was added. The mixture was stirred at room temperature for 1 day and then poured into water. The product was extracted into ether twice and the combined organic phases were washed with water three times and dried over MgSO$_4$. The solvent was removed under a stream of nitrogen and the residue (44.9 mg, 100%) was used directly in the next step without further purification.

### 5-amino-1-(3,5-dichloro-4-(4-chlorobenzoyl)benzyl)-1H-1,2,3-triazole-4-carboxyamide (CAI) (9)

To 2-cyanoacetamide (14.4 mg, 0.17 mmol) in ethanol (1 mL), 30% sodium methoxide in methanol (31.8 µL, 0.17 mmol) was added. The mixture was heated at reflux for 40 min and after cooling slightly **42** (44.9 mg, 0.13 mmol) in ethanol (1 mL) was added. The mixture was heated at reflux for 1.5 hr and after cooling the solvent was removed under a stream of nitrogen. The crude product was purified by column chromatography (19:1 DCM/MeOH) to afford the title compound as a yellow solid (20.4 mg, 36%). $^1$H NMR (400 MHz, Methanol-$d_4$) δ 7.76 (d, J = 8.6 Hz, 2H), 7.55 (d, J = 8.6 Hz, 2H), 7.37 (s, 2H), 5.49 (s, 2H). $^{13}$C NMR (101 MHz, MeOD) δ 192.32, 166.86, 146.75, 142.14, 141.02, 138.00, 135.23, 133.16, 132.10, 130.62, 128.44, 123.23, 54.80. LCMS m/z calc. for C$_{16}$H$_{13}$Cl$_3$N$_5$O$_2$ [MH]$^+$; 424.0, found; 424.1, $t_R$ = 2.80 min. HRMS (ESI-TOF) m/z calc. for C$_{17}$H$_{13}$Cl$_3$N$_4$O$_2$ [M+H]$^+$; 424.0129 found; 424.0117 and 445.9965 [M+Na]$^+$.

