## [Decision Letter]

**Acceptance summary:**

This article combines chemical and cell biology assays to demonstrate hitherto unappreciated effects of clinically employed inhibitors on disruption of mitochondrial functions. Specifically, the authors show that mubritinib (a HER2 inhibitor) and carboxyamidotriazole (a calcium channel blocker) also suppress mitochondrial complex I. These findings have broad implications for therapeutic effects and toxicity of not only mubritinib and carboxyamidotriazole, but also potentially other drugs that harbor similar chemical moieties.

**Decision letter after peer review:**

Thank you for submitting your article "Identification of a novel toxicophore in anti-cancer chemotherapeutics that targets mitochondrial respiratory complex I" for consideration by *eLife*. Your article has been reviewed by three peer reviewers, including Ivan Topisirovic as the Reviewing Editor and Reviewer #1, and the evaluation has been overseen by Philip Cole as the Senior Editor.

After the discussion between the reviewers, the Reviewing Editor has drafted this decision to help you prepare a revised submission. Albeit, general policy of *eLife* is that the authors should aim to send back revised versions of their articles within two months, considering current situation and restricted access to the labs around the world, we will be happy to see your revision whenever it is completed. So please take your time and stay safe.

Summary:

The authors provide evidence that anti-neoplastic agent mubritinib (thought to act as specific HER2 inhibitor) exerts its effects in large part by inhibiting mitochondrial complex I. Based on their structure-activity relationship studies, which also included a number of mubritinib derivatives, it was concluded that the effects on mitochondrial complex I are mediated by 1*H*-1,2,3-triazole ring. Same putative toxicophore was also identified in unrelated ant i-cancer drug carboxyamidotriazole. Based on this, it was suggested that 1*H*-1,2,3-triazole may represent a toxicophore which interferes with mitochondrial complex I function. These findings therefore may potentially have broad implication on therapeutic effects and toxicity of mubritinib, carboxyamidotriazole and potentially other drugs. Notwithstanding that these findings were found to be both important and exciting, several issues and concerns are brought to the authors' attention, whereby it was thought that they should be addressed to merit acceptance for publication of this work:

Essential revisions:

1) It was found that insufficient evidence was provided to support the tenet that 1*H*-1,2,3-triazole ring is responsible for the effects of the drugs on complex I. For instance, it was observed that compound 7, in which 1*H*-1,2,3-triazole is replaced with imidazole ring also potently represses complex I (e.g. Figures 2G and H). Similarly, the imidazole derivative (compound 12) appear to exhibit comparable or even more potent effects than carboxyamidotriazole (e.g. Figures 3B, C, and G). To this end, it was suggested that 1*H*-1,2,3-triazole ring binding to complex I may be influenced by other substructures of tested drugs and that on its own, 1*H*-1,2,3-triazole ring may not be critical for complex I inhibition. Doesn't imidazole finding somewhat undermine the central idea of the manuscript about the distinctive role of triazoles since many drugs also have imidazole groups?

2) The concentration of glucose in the media in the experiments where cardiomyocytes were supplemented with glucose or galactose should be indicated. Considering that data in Figure 1F-I and Figure 4A-F were normalised to vehicle it is not clear what is the effect of maintaining cells in galactose for 72h. These data should thus be included.

3) The interpretation of Seahorse experiments was found to be problematic since ECAR is not uniquely reflective of glycolysis and also depends on non-glycolytic extracellular acidification related to respiration (PMID: 26709455). Moreover, the authors should convert ECAR and OCR to the same units, compare the fraction of ATP produced by OXPHOS vs. glycolysis and assess bioenergetic capacity (PMID: 28270511). Seahorse data should also be normalised over DNA /protein content or cell number.

4) Several other issues were observed with the Seahorse data. Results in Figure 2B are incomplete as basal respiration in absence of FCCP is not presented. In Figure 2D, cells treated with rotenone have higher OCR than all other conditions, which is highly unexpected. What was the concentration of rotenone used? In Figure 3E, basal and maximal OCR should be presented on the same graph.

5) Experimental conditions were found to be inconsistent. For instance, analogous to the experiment in Figure 1 wherein mubritinib was compared tolapatinib, established Ca^2+^ channel inhibitor should be compared with carboxyamidotriazole. In Figure 3F, piericidin A was used as a complex I inhibitor, whereas rotenone was used in Figure 2D. In addition, PMP injection was included in Figure 2D but not in Figure 3F. In Figure 4A-F, mubritinib variant 5 was used as an inactive control, yet variant 6 was used for the cell index data (Figure 4G and Figure 1B). The authors should consolidate these experimental conditions to facilitate comparison of results from different experiments.

---

## [Author Response]

Essential revisions:1) It was found that insufficient evidence was provided to support the tenet that 1H-1,2,3-triazole ring is responsible for the effects of the drugs on complex I. For instance, it was observed that compound 7, in which 1H-1,2,3-triazole is replaced with imidazole ring also potently represses complex I (e.g. Figures 2G and H). Similarly, the imidazole derivative (compound 12) appear to exhibit comparable or even more potent effects than carboxyamidotriazole (e.g. Figures 3B, C, and G). To this end, it was suggested that 1H-1,2,3-triazole ring binding to complex I may be influenced by other substructures of tested drugs and that on its own, 1H-1,2,3-triazole ring may not be critical for complex I inhibition. Doesn't imidazole finding somewhat undermine the central idea of the manuscript about the distinctive role of triazoles since many drugs also have imidazole groups?

The starting point of the paper was an analysis of the anti-cancer therapeutic mubritinib, which contains a 1,2,3 triazole, and we then identified the same substructure in CAI. Therefore, the central point in the paper is that the 1,2,3 triazole, is the toxicophore within the context of these two anti-cancer drugs. This is stated in the Discussion “The toxicophore in the context of mubritinib and CAI is a 1*H*-1,2,3-triazole…”.

We agree with the reviewers about the importance of the imidazole group and our data clearly show that the 1,3 nitrogen linkage within imidazole is key since these compounds are active whereas the 1,2, nitrogen containing pyrizoles are inactive.

The importance of the 1,3 imidazole linkage was discussed throughout the manuscript

e.g.:

Results: “these data provide strong evidence that the 1,2,3-triazol-1-yl substituent or a 1,3-amidine like nitrogen atom relationship in the five-membered heterocycle, is required for complex I inhibition”;

Discussion: “and we have shown that the nitrogen at position 3 appears critical for both mitochondrial toxicity and efficacy”;

“the efficacy of both mubritinib and CAI (Figure 4) is dependent upon an amidine-like nitrogen substructure”.

Similarly, we have also discussed that IACS-010759 which is a 1,2,4 containing triazole also contains the same 1,3, nitrogen relationship (see the third paragraph of the Discussion). However, the role of the imidazole in terms of toxicity, does not undermine our central message, and data strongly suggest that the 1,3 imidazole within the 1*H*-1,2,3-triazole ring in CAI and mubritinib is critical for complex I inhibition.

To make it clearer we have modified the text as follows:

Abstract: “alters complex I inhibition, identifying the heterocyclic 1,3-nitrogen motif as the toxicophore”;

Introduction: “modifying the 1*H*-1,2,3-triazol-1-yl moiety present in mubritinib substantially alters both the inhibition of complex I and the toxicity to cardiomyocytes; identifying a heterocyclic 1,3-nitrogen motif as being key to….”;

Results: “These data provide strong evidence that a 1,3-amidine-like motif, housed within the 1*H*-1,2,3-triazol-1-yl substituent”;

“compounds which lacked the heterocyclic 1,3-nitrogen motif had no effect”;

“Taken together these data establish that the presence of the triazole and its embedded heterocyclic 1,3-nitrogen toxicophore is essential for the parent drug effects on tumour cell…”.

Additionally, and in support of our data, we have added a supplementary table which shows that Ketoconazole and Terconazole that only inhibit complex I contain 1,3 nitrogen linkage the same as CAI and mubritinib (see new Supplementary file 3) and text, see Discussion section “In a preliminary screen of compounds that only inhibited complex I…”

2) The concentration of glucose in the media in the experiments where cardiomyocytes were supplemented with glucose or galactose should be indicated.

hESC-CM cells were grown in RPMI 1640 media containing either glucose (11 mM) or galactose (10 mM). To make this clearer we have added this information to the figure legend and Materials and methods (subsection “Cell Culture”).

Considering that data in Figure 1F-I and Figure 4A-F were normalised to vehicle it is not clear what is the effect of maintaining cells in galactose for 72h. These data should thus be included.

There has been a slight misunderstanding of the data in these figures. Figure 1G and Figures 4A-F show the total number of live cells in the population and thus these values have not be normalised to a vehicle control sample. Importantly, when comparing cell death in untreated cells grown in either glucose or galactose containing media, these data show that all cell lines tested were viable in galactose media after 72 hours.

3) The interpretation of Seahorse experiments was found to be problematic since ECAR is not uniquely reflective of glycolysis and also depends on non-glycolytic extracellular acidification related to respiration (PMID: 26709455).

The reviewer has raised a valid point that ECAR is not unequally reflective of glycolysis. As this experiment is not directly related to the main conclusions of the paper, and to avoid further confusion, we have removed old Figure 2C from the manuscript.

Moreover, the authors should convert ECAR and OCR to the same units, compare the fraction of ATP produced by OXPHOS vs. glycolysis and assess bioenergetic capacity (PMID: 28270511). Seahorse data should also be normalised over DNA /protein content or cell number.

There has been a slight misunderstanding as to the rationale for using seahorse assays in this study. The Seahorse assay was used to validate and support the GLU/GAL phenotypic assays and screen compounds (mubritinib, carboxyamidotriazole, synthesised compounds variants) for prospective ETC inhibition. Although it would be interesting to dissect the fraction of ATP produced by OXPHOS and glycolysis in response to these treatments in the future, we feel that these questions are beyond the scope of this manuscript.

The same number of cells were seeded in each well with a multi-channel pipette and any well-to-well variation in seeding density in a single experiment (that was not been accounted for by three technical repeats) is mitigated by n=3/4 biological replicates that were performed on different days using separately passaged cells, and integrated into the statistical quantification. Crucially, we subsequently validate all cellular Seahorse assay data using the in vitro cell-free ETC biochemical assay.

4) Several other issues were observed with the Seahorse data. Results in Figure 2B are incomplete as basal respiration in absence of FCCP is not presented.

To make these data more complete we have added a representative seahorse trace for the data presented in Figure 2B, which shows basal respiration prior to FCCP addition (Figure 2—figure supplement 1A and B).

In Figure 2D, cells treated with rotenone have higher OCR than all other conditions, which is highly unexpected. What was the concentration of rotenone used?

The rotenone and untreated traces were unfortunately mislabelled. This has now been amended.

In Figure 3E, basal and maximal OCR should be presented on the same graph.

We have now combined basal and maximum OCR (old Figure 3Ei and 3Eii) into a single graph (Figure 3E), and we have also added representative seahorse traces to complement Figure 3C and E (Figure 3—figure supplement 1A, B and D).

5) Experimental conditions were found to be inconsistent. For instance, analogous to the experiment in Figure 1 wherein mubritinib was compared tolapatinib, established Ca^2+^ channel inhibitor should be compared with carboxyamidotriazole.

We have now included an analogous experiment to Figure 1B to compare signalling in A549 cells treated with CAI or a non-specific calcium channel inhibitor (bepridil hydrochloride) (Figure 3—figure supplement 3). Similar to mubritinib, CAI activates signalling pathways that are responsive to changes in energy status such as AMPK/ACC. Conversely, treatment with bepridil hydrochloride or the inactive CAI variant 11 did not enhance ACC phosphorylation, further suggesting that CAI is a complex I inhibitor and not a calcium channel inhibitor.

We have added additional text to describe this experiment to the last paragraph of the subsection “The 1,2,3-triazol-1-yl toxicophore in carboxyamidotriazole (CAI) inhibits ATP production, mitochondrial function and cell proliferation”.

In Figure 3F, piericidin A was used as a complex I inhibitor, whereas rotenone was used in Figure 2D.

Both Rotenone and Piericidin A, used in Figure 2D (now Figure 2C) and Figure 3F as control compounds are well established canonical complex I Q-site inhibitors and therefore they can be used interchangeably. What is central in terms of experimental output is that either compound specifically precludes complex I-driven oxygen consumption in the PMP assay, thus allowing the dissection of complex I activity from OCR that is driven by the remainder of the respiratory complexes (i.e. succinate/complex II). Assays with the different complex I inhibitors are configured to unequivocally validate these data; in Figure 2C (rotenone treatment) the collapsed OCR in permeabilised cells is resistant to Pyr/Mal injection whereas subsequent succinate injection increases OCR to untreated levels, and succinate-driven OCR is fully antimycinA-sensitive. The exact same assay control configuration is used to validate the complex I-specificity ofpiercidin A in the PMP assay (Figure 3F). Crucially, the data derived from the PMP assay is validated in the in vitro/biochemical assay; CAI and Mubritinib are specific complex I Q-site inhibitors.

In addition, PMP injection was included in Figure 2D but not in Figure 3F.

In Figure 2D (now Figure 2C), PMP was injected by the seahorse bioanalyser, as indicated within the figure, whereas PMP was added directly to the cells in Figure 3F immediately prior to starting the assay (i.e. at 0 hr). Although there was a difference in how PMP was used in terms of injection this is inconsequential in terms of the integrity of data and conclusions supported by this data; both approaches permeabilise cells, collapsing OCR and clearly confer cytosolic access to succinate and Pyr/Mal (they both potently stimulate respiration). The data derived from these different approaches/assays is appropriately controlled to show, and dissect, complex I-dependent OCR versus complex II-dependent OCR.

In Figure 4A-F, mubritinib variant 5 was used as an inactive control, yet variant 6 was used for the cell index data (Figure 4G and Figure 1B). The authors should consolidate these experimental conditions to facilitate comparison of results from different experiments.

To consolidate experimental conditions across the different experimental techniques we have made a number of modifications to these figures. We now show the effects of mubritinib variant 5 instead of variant 6 on AMPK/mTOR signalling in BT474 cells (Figure 1B and Figure 1—figure supplement 1C). Additionally, we have now included the proliferation of BT474 cells following treatment with mubritinib and compound variant 5 (Figure 4G).